# Risky Decision-Making in Adults with Alcohol Use Disorder—A Systematic and Meta-Analytic Review

**DOI:** 10.3390/jcm12082943

**Published:** 2023-04-18

**Authors:** Akke-Marij D. Ariesen, Julia H. Neubert, Geraldina F. Gaastra, Oliver Tucha, Janneke Koerts

**Affiliations:** 1Department of Clinical and Developmental Neuropsychology, University of Groningen, 9712 TS Groningen, The Netherlands; 2Department of Psychiatry and Psychotherapy, University Medical Center Rostock, 18147 Rostock, Germany; 3Department of Psychology, Maynooth University, National University of Ireland, W23 F2K8 Maynooth, Ireland

**Keywords:** alcohol use disorder, addiction, adults, risky decision-making, performance-based tasks

## Abstract

Alcohol use disorder (AUD) forms a major health concern and is the most common substance use disorder worldwide. The behavioural and cognitive deficits associated with AUD have often been related to impairments in risky decision-making. The aim of this study was to examine the magnitude and type of risky decision-making deficits of adults with AUD, as well as to explore the potential mechanisms behind these deficits. To this end, existing literature comparing risky decision-making task performance of an AUD group to a control group (CG) was systematically searched and analysed. A meta-analysis was performed to address overall effects. In total, 56 studies were included. In the majority of studies (i.e., 68%), the performance of the AUD group(s) deviated from the CG(s) on one or more of the adopted tasks, which was confirmed by a small to medium pooled effect size (Hedges’ *g* = 0.45). This review therefore provides evidence of increased risk taking in adults with AUD as compared to CGs. The increased risk taking may be due to deficits in affective and deliberative decision-making. Making use of ecologically valid tasks, future research should investigate whether risky decision-making deficits predate and/or are consequential to the addiction of adults with AUD.

## 1. Introduction

In everyday life, most people make decisions on a daily basis. Depending on the number of response options to choose from, and the immediate and future consequences associated with these options, these decisions vary in complexity [1]. For both simple and complex decisions, people often have to weigh the consequential risks and rewards of each response option to come to an adaptive choice [2]. This so-called ‘risky decision-making’ involves intuitive as well as deliberative thought processes [3,4]. Problems in this type of decision-making may lead to increased and unnecessary risk taking [2], which can negatively impact daily life. People with deficits in evaluating the potential risks and rewards of the choices they make may, for example, be at a heightened risk for engagement in criminal behaviour [5], risky sexual behaviour [6], and drug or alcohol abuse [7].

Alcohol use disorder (AUD) refers to a maladaptive pattern of alcohol intake characterized by the inability to stop or control alcohol consumption despite its detrimental consequences. AUD is accompanied by an increased alcohol tolerance, cravings, and withdrawal symptoms if the intake is halted [8]. With an estimated lifetime prevalence rate of 8.6% cross-nationally [9], AUD forms a major health concern and is the most common substance use disorder worldwide [10]. AUD is associated with various psychiatric comorbidities including mood, anxiety, and personality disorders [11]. Further, AUD has been related to cognitive impairments in the domains of executive functioning, processing speed, language, social cognition, visuospatial abilities, and memory and learning [12,13,14]. The chronic and excessive alcohol consumption as seen in AUD moreover puts individuals at risk of developing alcoholic Korsakoff’s syndrome. Korsakoff’s syndrome is a disorder primarily characterized by profound amnesia [15]. In the literature, the behavioural and cognitive deficits associated with AUD have often been related to deficits in risky decision-making. Abnormalities in decision-making have even been described a central feature of (alcohol) addiction [16,17,18]. Indeed, adults with AUD appear more likely to engage in (health-related) risk behaviour than healthy individuals [19,20,21]. Importantly, this increase in risk-taking behaviour could be both a consequence of and a risk factor for the maintenance of alcohol use [22].

An influential explanatory framework for the association between deficits in (risky) decision-making and AUD is the somatic marker theory (SMT) of addiction [23,24,25]. Central to the SMT is the idea that emotions guide the decision-making process [1]. The SMT states that deficiencies in the emotional or ‘somatic’ signalling of the prospective consequences of choice options lead to maladaptive response selection and increased risk-taking behaviour [1,25,26]. Accounting for deficits in intuitive or affective decision-making processes as well as in more cognitively demanding, deliberative decision-making processes [3,4], the SMT of addiction is largely compatible with the dual-process models of judgment and decision-making [24]. Specifically, the SMT proposes that the somatic signals that guide the decision-making process are brought about by two interacting neural systems. The ‘impulsive system’, typically associated with affective decision-making, responds to environmental stimuli indicative of immediate rewards or pleasure and activates feelings related to the immediate prospect of a decision such as the urge to obtain alcohol. The ‘reflective system’, on the other hand, is associated with deliberative decision-making, as it exerts a certain level of control over the impulsive system and can activate feelings related to the future prospects of a choice such as feelings of guilt following excessive alcohol consumption in the past [23,24,25].

The SMT postulates that during the decision-making process the somatic signals triggered by the impulsive and reflective system compete until one signal prevails. This signal consequently guides, or biases, the decision to be made [23]. In the context of this theory, alcohol addiction can thus be understood as an imbalance between the two systems. The addiction emerges either from a hyperactive impulsive system, where the rewarding impact of immediate alcohol consumption is overestimated, thus weakening the control of the reflective system, or from a dysfunctional reflective system, where adverse consequences of the decision to drink are disregarded, as feedback from prior experiences cannot be integrated in the brain [23,24,25]. In regulating the impulsive system, the reflective system is moreover thought to be particularly dependent on brain regions associated with executive functioning (e.g., regions of the prefrontal and cingulate cortex) [24]. Further adding to the link between risky decision-making deficits and AUD, excessive alcohol consumption has often been associated with global impairments in executive functioning [22,27]. Indeed, adults with AUD have been found to have impairments in various aspects of executive functioning relevant for deliberative decision-making including working memory, response inhibition, cognitive flexibility, and planning [13,14,28]. Importantly, these global impairments in executive functioning as well as the imbalance between the two systems as proposed by the SMT have been suggested to be both a vulnerability factor for the development of addiction and to be involved in the maintenance of alcohol use [22,24,25].

In line with the SMT of addiction and the hypothesized deficits in affective and deliberative decision-making processes, behavioural research has recurrently shown adults with AUD to have problems in various aspects of (risky) decision-making. Specifically, as compared to healthy individuals, adults with AUD were consistently found to engage in more impulsive choice behaviour and to show deficits in delay of gratification and delay discounting [29]. These deficits indicate that adults with AUD were more likely to prefer choices linked to smaller immediate rewards over choices linked to larger delayed rewards [30,31,32], as well as to disregard future losses in order to avoid immediate losses [33]. Various studies further indicated that adults with AUD may demonstrate a reduced aversion to losses [34] and risks [35] in the prospect of potential gain. As compared to healthy individuals, adults with AUD appear to have an aberrant sensitivity to feedback in the form of punishment and reward [36]. This altered sensitivity reduces the possibility to learn from previous decisions. Indeed, in a recent meta-analytic review on risky decision-making in people with substance use disorders, based on subgroup analysis, Chen et al. (2020) concluded that individuals with AUD showed increased risky decision-making on behavioural tasks as compared to non-using or limited use controls [7]. In the context of this study, it should be noted, however, that Chen et al. (2020) [7] addressed risky decision-making in individuals with substance use disorders in general, and adopted corresponding study aims (e.g., investigating the impact of (poly)substance dependency on task performance).

Whereas both theoretical and behavioural research thus seem to have established a link between increased risk taking and AUD, no extensive (meta-analytic) literature review has been performed to date that focuses on risky decision-making in adults with AUD, specifically. The objective of the present study is therefore to provide a comprehensive overview and meta-analysis of the existing studies that compare the performance on risky decision-making tasks of an AUD group with the performance of a control group (CG). This study therewith aims to examine the magnitude and type of deficits in risky decision-making of adults with AUD and to explore the potential mechanisms behind these deficits. By adding to our understanding of the link between risky decision-making and AUD, the outcomes of this study may prove useful for clinical practice. Specifically, a better understanding of this link can provide insight into the development and maintenance of alcohol addiction, which may aid in relapse prevention and the enhancement of treatment options for adults with AUD.

## 2. Method

### 2.1. Study Selection Procedure

A systematic search of the available literature addressing AUD and (risky) decision-making was carried out according to the guidelines of the Preferred Reporting Items for Systematic Reviews and Meta-Analyses (PRISMA; [37]) (PRISMA checklist; Appendix A). The present study was not pre-registered as the literature search and data extraction were already carried out before registration was considered. Therefore, no registration information is available for the protocol. Journal articles were searched using the databases PsycINFO, MEDLINE, PubMed, and Web of Science. A combination of the primary search terms associated with AUD (i.e., “alcohol use disorder”, alcoholism, alcohol abuse, alcohol dependen*, alcohol misuse, Wernicke*, or Korsako*) and the secondary search terms related to decision-making (i.e., decision making, decision-making, decision, judgment, or judging) had to appear in the title and/or the abstract of the articles. Only peer-reviewed articles that were written in English were included in this review. Further inclusion criteria were formulated using the Patient/Population, Intervention, Comparison, Outcomes (PICO) framework. This framework was slightly adapted in that a criterion was formulated to address the adopted assessment method rather than an intervention method. Specifically, studies were only included when they (a) included a group of adults with AUD as their main clinical diagnosis, (b) adopted one or more performance-based task used to assess risky decision-making, (c) compared the AUD group to a CG without (severe) psychiatric or neurological disorders, and (d) reported test statistics for group comparisons between the AUD group(s) and CG(s) on the behavioural outcome measure(s) of the risky decision-making task(s). The AUD group was considered to be clinically diagnosed if the participants were diagnosed according to DSM-5, DSM-IV-TR, DSM-IV, or ICD-10 criteria for AUD, alcohol abuse, or alcohol dependence. To ensure clarity and consistency, the term AUD will be used throughout this study while also referring to diagnoses of alcohol abuse and alcohol dependence, which have been combined into the overarching DSM-5 construct of AUD [38]. Both AUD groups without comorbidities (i.e., ‘pure’ AUD groups) and AUD groups with AUD as their main clinical diagnosis, but with comorbid psychiatric disorders, were included in this review.

The literature search for the systematic review was completed on the 24th of March 2023. After the search, duplicates were removed from the literature list. The retrieved literature was then supplemented with relevant literature cited in the articles found (manual search). Abstracts of the remaining studies were independently screened for eligibility for inclusion by one of two reviewers (JHN, ADA). After this initial screening, the remaining articles were read in full in order to identify which articles fulfilled the inclusion criteria. Any uncertainty or disagreement about study inclusion was resolved by discussion with a third reviewer (JK). The articles that fulfilled all criteria mentioned above were subsequently included in the review.

### 2.2. Study Analysis

#### 2.2.1. Content Analysis

A content analysis was conducted for the included studies. The results were independently analysed and extracted by ADA and JHN, including the following aspects: first author and year of publication, sample characteristics (i.e., demographic characteristics, comorbidities of the AUD group, and relevant exclusion criteria), characteristics related to the alcohol consumption of the AUD group (e.g., frequency, duration, and quantity of alcohol consumption, and abstinence period (before assessment)), adopted risky decision-making task(s), and the main study outcomes considered relevant for the research question at hand. Relevant study outcomes included between-group comparisons of the task performance of an AUD group and a CG or comparisons between the task performances of different AUD groups, as well as reported associations between outcome measures of the risky decision-making task(s) and demographic, clinical, or alcohol-use-related variables in the AUD group(s) and/or CG(s). Group differences were considered significant at the alpha levels used in the original studies. Furthermore, study outcomes regarding between-group comparisons of the task performance of the AUD group(s) and CG(s) were interpreted and categorised as (a) studies predominately finding significant between-group differences in the most important outcome measures related to risky decision-making, (b) studies predominantly finding non-significant differences between the AUD group and the CG in the most important outcome measures related to risky decision-making, or (c) studies finding mixed results in that group differences were observed in some, but not all outcome measures related to risky decision-making, or on some, but not all included risky decision-making tasks. To ensure correctly weighted conclusions, studies that most likely made use of the same or largely overlapping participant sample(s) as one of the other studies were considered conjointly (i.e., counted once) with regard to the overall conclusions on differences between the AUD groups and CGs in both the content and meta-analysis.

#### 2.2.2. Meta-Analysis

In addition to the content analysis, a meta-analysis was performed to determine to what extent adults with AUD show increased risk taking on decision-making tasks as compared to control participants without (severe) psychiatric or neurological disorders. Apart from fulfilling the inclusion criteria for the qualitative review as described above, the studies that were included in the meta-analysis had to (a) report on an outcome measure directly related to the level of risk taking of the participants (i.e., outcome measures such as ‘money gained’ and ‘loss aversion’ were not included) and (b) provide appropriate statistical data to calculate the effect size. In addition to a global meta-analysis, subgroup analyses were performed for the two decision-making tasks that were most frequently adopted in the included studies, i.e., the Iowa Gambling Task (IGT) and the Cambridge Gambling Task (CGT). The other risky decision-making tasks were only used in one to five of the included studies, which was considered too limited to perform relevant subgroup analyses.

With regard to the outcome measures related to risk taking that were used for the meta-analysis, the ‘net score’ or the ‘number of advantageous or disadvantageous choices’ were used for the IGT. For the CGT ‘risk taking’, the ‘overall proportion bet’, or the ‘quality of decision-making’ were used as risk-related outcome measures. Other outcome measures that were included in the global meta-analysis only were the ‘average (adjusted) number of pumps’ for the Balloon Analogue Risk Task (BART), the ‘number of risky choices’ for the Cups Task (CT), and the ‘risky responses divided by non-risky responses’ for the Risk-Taking Task (RTT). If a study provided results for multiple AUD groups and/or at different points in time, the AUD group with the least comorbidities (i.e., the ‘purest’ AUD group), the AUD group with merged separate groups (e.g., a total AUD group with females and males together), and/or the first time point was used for the meta-analysis. If more than one CG was included (e.g., a smoking and a non-smoking CG), the CG that matched the AUD group most in terms of non-alcohol-related variables was used as comparison group. A mean effect size across tasks was computed when studies reported on risk taking on two or more tasks, using a conservative approach by treating the two outcomes as independent [39] (p. 237). The effect size was (re)coded so that a positive effect size indicates higher risk taking in the AUD group as compared to the CG.

Hedges’ *g*, which is an adjustment of Cohen’s *d* for small sample sizes, was used as a measure of effect size [40]. The effect size was calculated from sample sizes, means, and standard deviations or, if this information was not available, derived from test statistics such as *F* and *p*-values [41]. Considering the heterogeneity in study characteristics, the random-effects pooling method was used to estimate the overall effect size across studies. An effect size in the order of 0.20, 0.50, and 0.80 can be interpreted as small, medium, and large, respectively [42]. Heterogeneity was assessed using the *I*^2^ statistic [43]. *I*^2^ values in the order of 25%, 50%, and 75% can be considered as low, moderate, and high, respectively. Publication bias was inspected visually using a funnel plot. The meta-analysis was performed with and without outliers, which were identified visually in the funnel plot. The meta-analysis was performed in Review Manager [44].

## 3. Results

The search resulted in a literature list of 4720 articles published between 1906 and 2023, from which duplicates were removed. Six articles were added to the retrieved literature based on a manual search. A total of 2253 studies were screened for eligibility, and 180 articles were read in full. A summary of the search and review process is presented in Figure 1. In total, 56 studies were included in this review. Upon further inspection, 12 of the included studies were found to most likely have made use of the same, or largely overlapping participant sample(s) as one of the other studies. This concerned the studies by Arcurio et al. (2015) and Folco et al. (2021) [20,45], Bjork et al. (2008) and Zhu et al. (2016) [46,47], Fishbein et al. (2007) and Flannery et al. (2007) [48,49], Galandra et al. (2021) and Canessa et al. (2021) [50,51], two studies by Loeber et al. (2009; 2010) [52,53], and two studies by Noël et al. (2007; 2011) [54,55]. These 12 studies were therefore considered conjointly (i.e., counted once) in the analyses provided below (i.e., numbers, percentages, and meta-analyses). In total, 50 studies presumably made use of distinct participant samples and were considered separately in the content analysis (see Figure 1).

For the global meta-analysis, five of the fifty-six included studies were excluded because they did not report on a risk-related outcome measure [34,35,56,57,58], and five studies were excluded because they did not provide appropriate statistical data [20,45,59,60,61]. Making use of overlapping participant samples, two studies were furthermore excluded from the meta-analysis because the statistical data provided and the textual description of the data were not in line with each other [46,47]. In total, 44 studies could therefore be included in the global meta-analysis, 40 of which presumably made use of distinct participant samples, and were therefore considered separately (see Figure 1).

**Figure 1 jcm-12-02943-f001:**
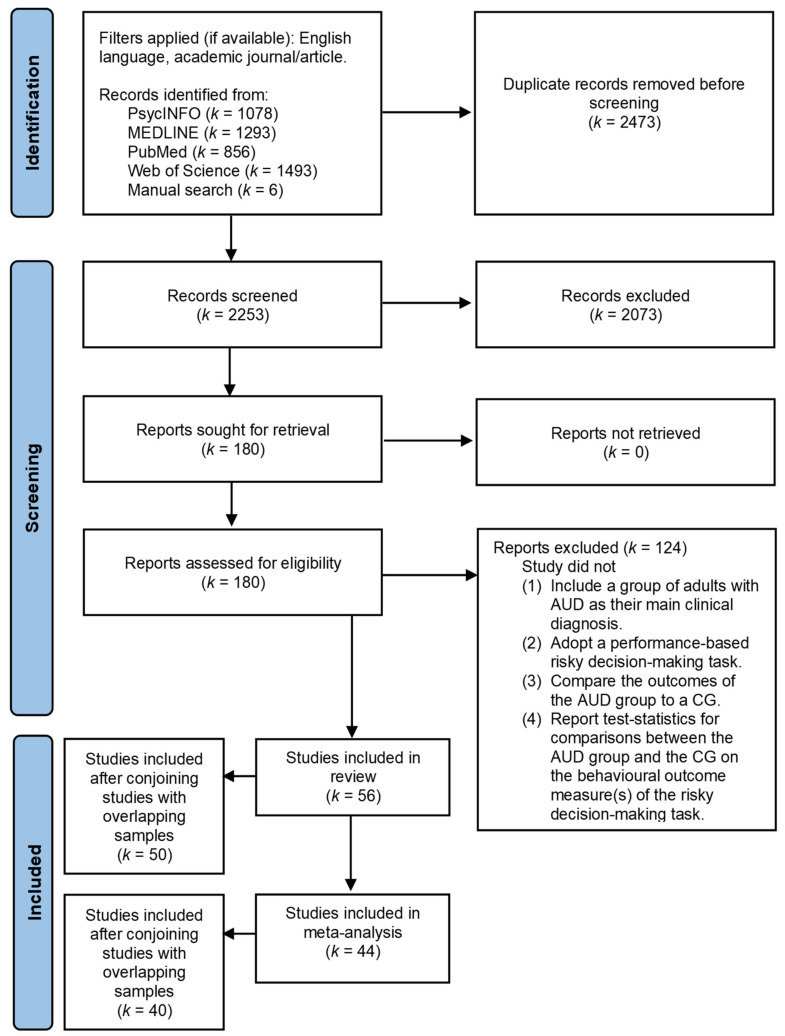
Flow diagram of the systematic search and review process according to the guidelines of Preferred Reporting Items for Systematic Reviews and Meta-Analyses (PRISMA 2020; [37]).

### 3.1. Overall Task Findings

All studies adopted one or more performance-based tasks to assess risky decision-making in adults with AUD. A total of 30 of the 50 studies made use of the IGT to assess risk-taking behaviour. One study additionally made use of a variant version of the IGT. Other risky decision-making tasks that were used in two or more of the included studies were the CGT, the BART, the RTT, the Game of Dice Task (GDT), and the Coin Flipping Task/Loss Aversion (Gambling) Task (CFT/LA(G)T). The Ecological Decision-Making Task (EDMT), Explicit Gambling Task (EGT), CT, Card Playing Task (CPT), Wheel of Fortune (WoF) task, Single Outcome Gambling (SOG) task, Lane Risk-Taking Task (LRT), Probabilistic Discounting Task (PDT), and the Mixed Gambles Task (MGT) were used in one study only. A description of the identified risky decision-making tasks is shown in Table 1. Table 2 provides a comprehensive overview of the characteristics and main outcomes of the included studies.

Overall, 34 of the 50 studies (68%) reported that the AUD group(s) showed an aberrant performance as compared to the CG(s) on one or more of the outcome measures of the adopted risky decision-making task(s), indicating increased risk taking in adults with AUD. In 15 of the 50 studies (30%), no significant group differences in risky decision-making were found between the AUD groups and CGs. Finally, in one study (2%), outcomes on one of the two adopted tasks were indicative of reduced risk taking in adults with AUD as compared to the CG, whilst on the other task, no significant between-group differences were observed [62]. As described above, 40 studies were included in the global meta-analysis (see Figure 2). The pooled effect size of these studies was 0.45 (95% CI = 0.35–0.55, *p* < 0.001; small to medium effect), which corresponds to the finding that adults with AUD showed significantly increased risk taking on the decision-making tasks as compared to the CGs. Furthermore, heterogeneity of the studies included in the meta-analysis was low to moderate (*I*^2^ = 45%, *p* = 0.001). As shown in Figure 3, the funnel plot shows no clear asymmetry, suggesting the absence of a publication bias. Finally, there is one data point that may be considered as an outlier. Removing this outlier did not significantly influence the outcome of the meta-analysis (pooled Hedges’ *g* = 0.43, 95% CI = 0.34–0.52, *p* < 0.001).

**Table 1 jcm-12-02943-t001:** Overview and description of identified risky decision-making tasks.

Task Name	Task Description	Outcome Measures
Iowa Gambling Task (IGT) [63]	▪ At the start of the task, the participant receives a starting capital and is instructed to maximize their gain. ▪ The participant chooses 1 card from 4 decks of cards 100 times. ▪ For each card drawn, (hypothetical) money is gained/lost. ▪ A total of 2 out of the 4 decks are considered advantageous/safe (i.e., consistent selection from these decks leads to a net gain). ▪ A total of 2 out of 4 decks are considered disadvantageous/risky (i.e., consistent selection from these decks leads to a net loss). Variant version used in Kim et al. (2006) [64]: ▪ The order of punishment and reward of the four decks is reversed as compared to the original version. The two tasks are otherwise similar in operation and appearance: ▪ A total of 2 out of the 4 decks are associated with high immediate punishment, but with higher future rewards (i.e., advantageous decks). ▪ A total of 2 out of 4 decks are associated with low immediate punishment, but with lower future rewards (i.e., disadvantageous decks).	Scores are usually calculated for five blocks of 20 trials each and for the entire task (all 5 blocks combined): ▪ Number of choices from each deck. ▪ Number of safe/advantageous choices. ▪ Number of risky/disadvantageous choices. ▪ Net score (i.e., total number of safe choices minus total number of risky choices). ▪ Financial gain and loss. ▪ Total financial outcome. ▪ Conceptual knowledge/strategy insight (i.e., identification of (dis)advantageous decks at the end of the task).
Cambridge Gambling Task (CGT) [65]	▪ At the start of the task, the participant is told that a yellow token has been hidden inside 1 box out of a row of 10 boxes (some boxes are red, others blue). ▪ The participant is instructed to maximize their total number of points earned by correctly indicating whether the token is in a red or in a blue box. ▪ The participant is then offered a series of betting options to place bets on whether their choice (red or blue box) is correct. ▪ The chosen bet is added/subtracted from the total number of points earned depending on whether the choice made was correct or incorrect. ▪ The task is usually performed in 2 conditions (containing 4 blocks of 9 trials each): Ascending condition: the series of offered bets starts small and increases. Descending condition: the series of offered bets starts large and decreases.	▪ Quality of decision-making (i.e., percentage of trials in which the more likely outcome was chosen). ▪ Overall proportion of bets made (i.e., proportion of the total points placed on a choice, when that choice was the more likely outcome). ▪ Sum of bets (i.e., average bet across blue to red box ratios). ▪ Risk adjustment (i.e., the degree to which the number of points put at risk by the participant increases when the ratio of blue to red boxes becomes more favourable). ▪ Risk taking (i.e., mean proportion of the current points total that the participant chooses to risk on trials for which they chose the more likely outcome). ▪ Delay discounting (i.e., preference for smaller, immediate rewards over larger, delayed rewards). ▪ Bankruptcies (i.e., trials in which the participant loses all points within one block). ▪ Deliberation time (i.e., speed of decision-making).
Balloon Analogue Risk Task (BART) [66]	▪ At the start of the task the participant is instructed to maximize their gain by inflating virtual balloons. ▪ For each pump of air, the participant can earn money, but each pump of air also increases the risk that the balloon will pop. ▪ When the balloon pops, all money earned during a trial is lost, and a new trial begins. ▪ In each trial, the participant has the option to click the ‘collect’ button, enabling them to end the trial at any time and collect the money earned before the balloon pops (i.e., cash-out decisions).	▪ Adjusted number of pumps (i.e., average number of pumps based on the trials where the balloon did not pop). ▪ Number of balloons that popped. ▪ Total financial outcome.
Risk-Taking Task (RTT) [67]	▪ At the start of the task, the participant is instructed that they will receive money for the points that they earn during the task. ▪ In each of the task trials, the participant can either earn points by repeatedly clicking on a ‘GO’ button or end the trial by clicking on a ‘STOP’ button. ▪ The task consists of two consecutive conditions: Green background: in this phase every click on the ‘GO’ button advances the total trial score by two points, and there is no risk of losing points. Yellow background: in this phase every click on the ‘GO’ button advances the total score with two more points than the number of points that were earned for the previous response (i.e., successive ‘GO’ responses increase the reward). ▪ The participant is instructed that in order to keep the money earned, they must click the ‘STOP’ button before the screen turns from yellow to red. They, however, do not know for how long the screen will be yellow. ▪ If the participant clicks the ‘STOP’ button when the screen still is yellow, the screen turns blue, and the participant earns the points. ▪ If the participant clicks the ‘STOP’ button after the screen turned red, the participant earns no points in the trial. ▪ A continued ‘GO’ response in this condition is thus associated with greater reward (i.e., more points/money can be earned) as well as greater risk (i.e., the trial could be terminated before the participant clicks ‘STOP’). ▪ The task is designed in such a way that the optimal strategy entails taking some risk. Adaptations in Bjork et al. (2008)/Zhu et al. (2016) [46,47]: ▪ A third, high-penalty condition is added. In this condition, points are deducted from the total points earned when the participant clicks the ‘STOP’ button too late (i.e., after a ‘bust’). ▪ Contrastingly, no points can be lost in the green background condition (no-penalty condition), and in the yellow-background condition no points are earned after a bust, though no points are deducted either (low-risk condition).	▪ Total financial outcome. ▪ Total number of ‘busts’ (i.e., number of times the screen turned red before clicking ‘STOP’). ▪ Trial average risky responding (i.e., ratio of the number of yellow-screen rewarded responses divided by the number of green-screen rewarded responses across all trials). ▪ Trial maximum risky responding (i.e., ratio of the maximum number of yellow-screen rewarded responses emitted in a trial divided by the number of green-screen rewarded responses). ▪ Risk taking (i.e., mean reward accrual time in non-busted trials).
Game of Dice Task (GDT) [68]	▪ At the start of the task, the participant receives a starting capital and is instructed to maximize their gain by betting on the value of a virtual die (or several dice). ▪ After every throw of the die/dice, money is gained/lost depending on whether the value corresponds to the bet made by the participant. ▪ Bets can be placed on the outcome of a single die or on combinations of outcomes of up to four dice. ▪ The choices for bets are related to stable and inversely proportional gains/losses and winning probabilities. ▪ Choices related to small gains/losses and a winning probability of >50% are considered ‘safe choices’ (i.e., betting on the value of 3 or 4 dice). ▪ Choices related to large gains/losses and a winning probability of <50% are considered ‘risky choices’ (i.e., betting on the value of 1 or 2 dice).	▪ Number of safe choices. ▪ Number of risky choices. ▪ Total financial outcome. ▪ Net score (i.e., total number of safe choices minus total number of risky choices).
Coin Flipping Task (CFT)/Loss Aversion (Gambling) Task (LA(G)T) ^a^ [69]	▪ During the task, participants are asked to decide whether they would accept or reject mixed gambles (i.e., gambles associated with a certain loss as a well as a certain gain) that offered a 50/50 chance of gaining an amount of money or losing another amount. ▪ After the participant has responded to the gamble with ‘accept’ or ‘reject’ they are asked to indicate the level of acceptance of the gamble (e.g., ‘strongly accept’, ‘weakly accept’, ‘weakly reject’, ‘strongly reject’). ▪ Possible gambles consist of various levels of gains and losses (e.g., in the study by Brevers et al. (2014) [70], the sizes of the potential gains ranged from $10 to $40, and the sizes of potential losses ranged from $5 to $20).	▪ Behavioural loss aversion (i.e., ratio of the loss response to the gain response). ▪ Behavioural gain/loss sensitivity. ▪ Level of acceptance of the gamble/risk acceptance.
Explicit Gambling Task (EGT) [58]	▪ At the start of the task, the participant receives a starting capital and is instructed to maximize their gain. ▪ The participant chooses 1 card from 4 decks of cards 45 times. ▪ When a card is turned, the chosen card displays a mention of either ‘positive value: gain’, ‘negative value: loss’ or ‘blank: neither gain nor loss’. ▪ After each card selection, the participant has to place the card below its card deck openly (i.e., showing the outcome value), enabling the participant to see the values of all selected cards from the four decks. ▪ After each card selection, the participant additionally receives immediate feedback of gain or loss (i.e., the participant is handed play currency in accordance with the amount gained/has to return play currency in accordance with the amount lost). ▪ Deck A is associated with smaller wins and losses than deck B over time, deck B is associated with smaller wins and losses than deck C over time, and deck C is associated with smaller wins and losses than deck D over time. ▪ In the long run, decks A and B are considered more advantageous than decks C and D.	▪ Number of choices from each deck. ▪ Total financial outcome.
Cups Task (CT) [71]	▪ This task consists of two conditions: Gain domain: the participant is asked to choose between a certain small gain and a gamble that results either in a larger gain or no gain. Loss domain: the participant is asked to choose between a certain small loss and a gamble that results either in a larger loss or no loss. ▪ For both conditions, the selection of the certain small gain or loss is considered the safe option and the gamble is considered the risky option. ▪ The probabilities for the larger wins or losses (0.20, 0.33, 0.50) and the size of possible wins or losses vary between trials so that the expected value for the risky option shifts from more favourable to less favourable. ▪ At the start of each trial an array of 2, 3, or 5 cups is shown on one side of the screen along with the possible gain or loss. ▪ After the participant made their choice between the risky option (i.e., the selection of one of the cups from the total amount of cups in the array leads to a designated amount of money gained/lost, whereas the other cups lead to no gain) or the safe option (i.e., the certain win/loss), the result of their choice (risky or safe) is directly revealed.	▪ Number of risky choices made in the gain and loss conditions at three different expected value levels (i.e., risk-advantageous, risk-equal, and risk-disadvantageous expected value (see [70])
Card Playing Task (CPT) (adapted from [72], see [73])	▪ At the start of the task, the participant is instructed to maximize their gain by playing cards from a deck in which face and number cards represent gains and losses of 0.50, respectively. ▪ The task consists of 10 blocks of 10 cards per block. ▪ Unbeknownst to the participant, the ratio of wins to losses changes with each task block. Per block, the number of cards increases with one loss card and decreases with one win card (i.e., in the first block the win/loss ratio is 9:1, in the second block 8:2, etc.) ▪ After completing each block, the participant is asked to decide whether to continue the task or to quit playing.	▪ Total number of cards played. ▪ Total financial outcome.
Wheel of Fortune (WoF) Task (adapted from [74], see [56])	▪ In this task, the participant is asked to select one of two gambles, represented by two wheels of fortune. ▪ Each wheel of fortune is divided into a red area, representing a loss, and a green area, representing a gain. ▪ The potential outcomes of each wheel of fortune are depicted next to the wheel and include a pair of gains (200 or 50) and losses (−200 and −50). ▪ The respective size of the wheel of fortune’s red and green area represent the different probability levels (20–80, 50–50, 80–20) of the outcomes. ▪ After the wheels of fortune are presented and the participant made a choice between the two gambles, the outcome of their choice is revealed (i.e., whether they won/lost). ▪ The task consists of two conditions: Partial feedback condition: the participant only is able to see the outcome of the wheel of fortune they selected. Complete feedback condition: the participant is able to see the outcomes of both wheels of fortune. ▪ After the outcome of their choice is revealed, the participant is asked to indicate their feelings regarding the outcome on a scale ranging from 50 (extremely elated) to −50 (extremely disappointed).	Scores can be calculated per feedback condition: ▪ Total financial outcome. ▪ Number of time-outs (i.e., trials in which the participant failed to respond within a predefined timeframe). ▪ Response times.
Single Outcome Gambling (SOG) task [57]	▪ At the start of the task, the participant is instructed to maximize their gain by selecting either the number ‘10’ or the number ‘50’, representing a monetary value of 10 cents or 50 cents, respectively. ▪ After the participant has selected one of the numbers in a trial, the chosen number either appears in a green box, indicating the chosen value was gained, or in a red box, indicating the chosen value was lost. ▪ Unbeknownst to the participant, the probability of wins/losses is 50/50, and the wins and losses are pseudorandomized.	▪ Selection frequency of the ‘10’ and ‘50’ options after a single ‘loss trial’ (i.e., −10 or −50) or ‘gain trial’(i.e., +10 or +50). ▪ Selection frequency of the ‘10’ and ‘50’ options after a ‘loss trend’ or ‘gain trend’ of the previous two, three, or four trials (i.e., after two, three or four consecutive ‘loss trials’ or ‘win trials’). ▪ Reaction times.
Lane Risk-Taking Task (LRT) (adapted from [75], see [59])	▪ At the start of the task, the participant receives a starting capital of $5 and is instructed to maximize their gain. ▪ At the beginning of each trial, two white squares are presented briefly, and the participant is asked to choose one of the two squares. ▪ One square is presented with a question mark underneath; this is considered the ‘risky’ square, as a choice for this square may result in a gain of $1 or $5, but may also result in a loss of −$1 or −$5. ▪ The other square is presented without a question mark and is considered the ‘safe’ square as a choice for this square results in a guaranteed win of $0.25. ▪ Unbeknownst to the participant, the ratio of wins to losses of the ‘risky squares’ is 50/50, and wins and losses are pseudorandomized. ▪ Consistent selection of the ‘safe square’ will result in a net gain of $10, and consistent selection of the ‘risky square’ will result in a net gain of $0 or $35.	▪ Number of ‘safe square’ selections. ▪ Number of ‘risky square’ selections.
Probability Discounting Task (PDT) (adaptation of existing task paradigm, see [35])	▪ At the start of the task the participant receives a starting capital, and in each trial is instructed to choose between two offers that appear simultaneously on the screen. ▪ In each trial, the participant is on the one hand offered smaller certain outcomes (i.e., gains or losses) and on the other larger probabilistic outcomes (i.e., gains or losses), where the size and probability of receiving the gain or loss are varied across trials. ▪ When the participant has made a decision between the two offers, the selected offer is highlighted in a red frame, and the next offer is presented. ▪ Possible gains range from €0.30 to €10, possible losses range from €0.30 to €−10. ▪ Possible probability values of the gains and losses are 2/3, 1/2, 1/3, 1/4, and 1/5. ▪ The participant does not receive feedback about the outcomes of their choices during the task.	▪ Risk-aversion regarding gains (i.e., preference for certain gains over probabilistic gains). ▪ Risk-seeking regarding losses (i.e., preference for probabilistic losses over certain losses).
Mixed-Gambles Task (MGT) (adaptation of existing task paradigm, see [35])	▪ At the start of the task, the participant receives a starting capital, and in each trial is instructed to choose between two offers that appear simultaneously on the screen. ▪ In each trial, the participant can choose between an offer of an unknown outcome (presented as an ‘x’) and an offer consisting of two possible known outcomes; a gain and a loss (e.g., ‘15’ and ‘−8’). ▪ When the participant has made a decision, the selected offer is highlighted in a red frame, and the next offer is presented. ▪ Possible gains range from €1 to €40, possible losses range from €−5 to €−20. ▪ The gain to loss ratio of the gambles is 50/50. ▪ The participant does not receive feedback about the outcomes of their choices during the task.	▪ Loss aversion (i.e., the ratio of the contribution of the loss magnitude and that of the gain magnitude to the subject’s decision; high values reflect a higher degree of loss aversion).
Ecological Decision-Making Task (EDMT) [45] (see also [20])	▪ During this task, pictures are presented to the participant that fall into one of four stimuli categories: alcoholic beverages, food, household/stationary items, and male faces. ▪ The alcohol and food stimuli represent appetitive decision cues, the household items the neutral decision cues, and the male faces the sexual decision cues. ▪ Each of the stimuli is presented for 4 s along with the word ‘yes’ or ‘no’, and a number, which represents the risk information/context, creating a low and a high-risk context for each cue category. Alcohol cues: yes/no refers to whether or not the participant has a designated driver, and the number (1–6) refers to how many alcohol units the drink in the picture contains. Food cues: yes/no refers to whether or not the food establishment passed the health and safety inspection, and the number (200–800) refers to the caloric content of the food in the picture. Item cues: yes/no refers to whether or not the store has a return policy, and the number (2–20) refers to the costs of the item in the picture. Face cues: yes/no refers to whether or not the male usually uses condoms, and the number (2–8) refers to the number of sexual partners of the male in the picture. ▪ The participant is asked to appraise the cue and the risk information and rate their likelihood to drink the alcohol, eat the food, buy the item, or have sex with the person on a scale of 1 (very unlikely) to 4 (very likely).	▪ Likelihood of endorsement of high- and low-risk stimuli in each category (i.e., alcohol, food, items, faces). ▪ Reaction times for high- and low-risk stimuli in each category (i.e., alcohol, food, items, faces).

**Note:** ^a^ First introduced by Tom et al. (2007) [69], this task paradigm has no official name. In the study by Brevers et al. (2014) [70], the task is referred to as the Coin Flipping Task (CFT); in the study by Genauck et al. (2017) [34], it is referred to as the Loss Aversion Task (LAT); and in the study by Zorick et al. (2022) [62]; it is referred to as the Loss Aversion Gambling Task (LAGT).

**Table 2 jcm-12-02943-t002:** Overview of included studies on risky decision-making in adults with AUD.

First Author (Year)	Sample Characteristics ^1^	Comorbidities and Relevant Exclusion Criteria ^2^	Alcohol-Use-Related Variables ^3^	Abstinence Period ^3^	Risky Decision-Making Task	Main Study Outcomes ^4^	Conclusion ^5^
Arcurio et al. (2015); Folco et al. (2021) [20,45]	**AUD**^#^ (*n* = 15) Age (y): 21.20 ± 2.08 Sex: Female (all participants) Education: High school graduate: 20% Some college: 60% College graduate: 20%	Depression (BDI score): AUD = CG Exclusion criteria for both participant groups included current treatment for depression or anxiety, self-reported symptoms of psychosis or of TBI, and dependency of stimulants or marijuana.	Frequency (days/week): 4.20 ± 1.15 Quantity (drinks/week): 36.57 ± 18.10	Minimum (h): 24	EDMT	Arcurio et al. (2015) [45]: ▪ Endorsement of high-risk alcohol stimuli: AUD > CG ▪ Endorsement of low-risk alcohol stimuli: AUD = CG ▪ In both groups (AUD and CG), the rated likelihood to drink alcohol was significantly reduced in the low-risk condition compared to the high-risk condition. ▪ Endorsement of high- and low-risk food stimuli: AUD = CG ▪ Endorsement of high- and low-risk household item stimuli: AUD = CG ▪ Across all stimulus categories, high-risk stimuli were less frequently endorsed than low-risk stimuli in both groups (AUD and CG). ▪ In the AUD group, participants took a significantly longer time to make high-risk alcohol decisions compared to low-risk alcohol decisions. ▪ Difference in reaction time between high- and low-risk alcohol decisions: AUD > CG (marginally significant) ▪ In both groups (AUD and CG), participants took a longer time to respond to high-risk compared to low-risk alcohol and food stimuli, and a longer time to respond to low-risk compared to high-risk household item stimuli. Folco et al. (2021) [20]: ▪ Endorsement of high-risk sexual stimuli (faces): AUD > CG ▪ Endorsement of low-risk sexual stimuli (faces): AUD = CG ▪ High-risk stimuli were significantly less frequently endorsed than low-risk stimuli in both groups (AUD and CG). ▪ The AUD group showed less reduction than the CG in the rated likelihood to have sex with the person in the picture in the low-risk condition compared to the high-risk condition. ▪ Reaction times did not differ significantly across groups (AUD and CG), stimuli categories (sexual, food, household items), and risk conditions (high, low).	+
	**CG** (*n* = 16) Age (y): 20.25 ± 1.57 Sex: Female (all participants) Education: High school graduate: 6.3% Some college: 81.3% College graduate: 12.5%						
Avcu Meriç et al. (2022) [76]	**AUD*** (*n* = 52) Age (y): 45.27 ± 10.02 Sex: Male (all participants) Education: 10.04 ± 3.21	Exclusion criteria for the AUD group included having drug-related cognitive deficiencies, severe comorbid psychiatric disorders, or other medical conditions that could hamper the understanding of study instructions.	AUD duration (y): 10.44 ± 7.41 Dangerousness of alcohol consumption (AUDIT score): 27.27 ± 7.57 Alcohol craving (PACS score): 8.62 ± 8.71	Minimum (weeks): 3	IGT	▪ Total net score: AUD < CG Associations: ▪ In the AUD group, IGT net scores correlated significantly with alcohol craving scores on the PACS (negative correlation), and the PACS scores added significantly to the prediction of IGT performance in a multiple regression analysis. IGT net-scores did not correlate significantly with age, education, smoking amount or history, dangerousness of alcohol consumption (AUDIT score), or AUD duration. ▪ In the CG, IGT net scores did not correlate significantly with age, education, smoking amount or history, or dangerousness of alcohol consumption (AUDIT score).	+
	**CG** (*n* = 52) Age (y): 45.90 ± 11.71 Sex: Male (all participants) Education: 10.12 ± 3.87						
Bernhardt et al. (2017) [35]—study 2	**AUD^#^** (*n* = 114) Age (y): 44.77 ± 10.56 Sex (m/f): 96/18 Education: n.r. *Subdivided by relapse status to heavy-drinking during 48-week follow-up interval:* **AUD^#^ abstaining** (*n* = 27) Age (y): 44.14 ± 13.06 Sex (m/f): 20/7 Education: n.r. **AUD^#^ relapsing** (*n* = 58) Age (y): 46.03 ± 10.22 Sex (m/f): 51/7 Education: n.r.	Anxiety and depression (HADS scores): AUD > CG AUD abstaining = AUD relapsing Exclusion criteria for both participant groups included a history of or current neurologic or mental disorder.	AUD severity (ADS score): AUD: 14.69 ± 6.74 AUD abstaining: 14.67 ± 7.06 AUD relapsing: 15 ± 6.14 Alcohol craving (OCDS score): AUD: 11.81 ± 8.36 AUD abstaining: 11.27 ± 8.04 AUD relapsing: 10.95 ± 7.61 Alcohol consumption in past year (grams of alcohol/drinking occasion): AUD: 206.92 ± 125.94 AUD abstaining: 190.33 ± 96.28 AUD relapsing: 206.53 ± 105.50 Alcohol intake per binge-drinking event in past year (grams of alcohol): AUD: 276.95 ± 157.37 AUD abstaining: 258.67 ± 112.23 AUD relapsing: 290.02 ± 155.17 Cumulated lifetime alcohol intake (kilograms of alcohol): AUD: 1749.09 ± 1096.02 AUD abstaining: 1677.32 ± 1207.02 AUD relapsing: 1893.29 ± 1139.54	Days: 17 ± 10 (range: 4–50)	PDT MGT	▪ PDT risk aversion regarding gains: AUD < CG ▪ PDT risk seeking regarding losses: AUD < CG ▪ MGT loss aversion: AUD < CG ▪ PDT risk aversion regarding gains: AUD relapsing = AUD abstaining ▪ PDT risk seeking regarding losses: AUD relapsing < AUD abstaining ▪ MGT loss aversion: AUD relapsing = AUD abstaining Associations: ▪ First regression model: PD for losses, PD for gains, MG and a measure for delay discounting were entered as potential predictors for relapse to heavy drinking during the 48-week follow-up. The overall model was significant and PD for losses and delay discounting were found to be significant predictors for relapse. ▪ Second regression model: HADS, ADS, and OCDS scores were added as potential predictors. In this model, PD for losses was still a significant predictor, but delay discounting was no longer significantly associated with relapse. ▪ In the AUD group, PD for gains correlated significantly (negative correlation) with smoking status (i.e., smoking or non-smoking). No significant correlations were found between PD for gains and the alcohol-use-related measures (i.e., ADS and OCDS scores, alcohol consumption in the past year, alcohol intake per binge-drinking event in the past year, cumulated lifetime alcohol intake). PD for losses and the MGT did not correlate significantly with smoking status or any of the alcohol-use-related measures. ▪ In the CG, PD for gains or losses or the MGT did not correlate significantly with smoking status and any of the alcohol-use-related measures.	+
	**CG** (*n* = 98) Age (y): 43.75 ± 10.86 Sex (m/f): 81/17 Education: n.r.						
Bjork et al. (2004) [67]	**AUD^#^** (*n* = 130) Age (y): 39.8 ± 8.0 Sex (m/f): 96/34 Education (y): 13.6 ± 2.4	AUD: ≥1 mood disorder according to DSM-4 criteria: 80% ≥1 anxiety disorder according to DSM-4 criteria: 40% Exclusion criteria for the AUD group included a history of seizures, features suggestive of foetal alcohol syndrome or other neurologic disorders, or a presentation of psychotic symptoms.	n.r.	Minimum (days): 7	RTT	▪ Total financial outcome: AUD = CG ▪ Total number of ‘busts’: AUD > CG ▪ Trial average (risky/non-risky responses): AUD = CG ▪ Trial maximum (risky/non-risky responses): AUD > CG Controlled for education (education entered as covariate): ▪ Total number of ‘busts’: AUD > CG ▪ Trial average (risky/non-risky responses): AUD > CG ▪ Trial maximum (risky/non-risky responses): AUD > CG Associations: ▪ Education level independently correlated significantly with trial average and trial maximum (positive correlations) while controlling for group (AUD and CG). ▪ In the CG, age did not correlate significantly with any of the RTT outcome measures. In the AUD group, there was a significant negative correlation between age and RTT trial maximum. ▪ In the group of male AUD participants *(n* = 96), age of onset of heavy drinking did not correlate significantly with any of the RTT outcome measures while controlling for educational level.	+/−
	**CG** (*n* = 41) Age (y): 38.5 ± 11.6 Sex (m/f): 27/14 Education (y): 16.9 ± 3.0						
Bjork et al. (2008) [46]; Zhu et al. (2016) [47] The majority of participants from Zhu et al. (2016) [47] were analysed in Bjork et al. (2008) [46], with the addition of 17 control participants.	Bjork et al. (2008) [46]: **AUD^#^** (*n* = 17) Age (y): 32.9 (SD n.r.) Sex (m/f): 10/7 Education: n.r.	Bjork et al. (2008) [46]: Lifetime history of cocaine dependence: *n* = 16 Lifetime history of cocaine abuse: *n* = 1 Lifetime history of cannabis dependence: *n* = 10 Lifetime history of cannabis abuse: *n* = 1 Exclusion criteria for the AUD group included a history of seizures, psychosis or features indicative of foetal alcohol syndrome.	Zhu et al. (2016) [47]: Duration of heavy drinking (y): 10.5 ± 6.5	Minimum (days): 7	RTT	Bjork et al. (2008) [46]: ▪ Money earned during task: AUD = CG ▪ Latency to respond to ‘$’ cue: AUD = CG ▪ Mean reward accrual time in non-busted low-penalty trials: AUD < CG ▪ Across time, risk-taking in the low-penalty trials increased in AUD, but decreased in CG (significant effect). ▪ Mean reward accrual time (risk-taking measure) adjusted for NEO-neuroticism scores: AUD = CG ▪ Number of busts in low-penalty trials: AUD < CG ▪ Mean reward accrual time in non-busted high-penalty trials: AUD = CG ▪ Risk-taking across time in high-penalty trials: AUD = CG ▪ Number of busts in high-penalty trials: AUD = CG ▪ CG stopped reward accrual significantly sooner in high than in low-penalty trials, whilst AUD participants did not, indicating an effect of penalty size on risk-taking in CG but not in AUD participants. ▪ In both groups (AUD and CG), previous trial outcome (win or bust) significantly affected risk exposure time (i.e., shorter risk exposure after a bust) in low-penalty trials. In high-penalty trials there was no significant effect of previous trial outcome. ▪ Self-reported anxiety ratings reflected the magnitude of potential reward and penalty in CG but not in AUD participants. ▪ Self-reported boredom ratings reflected the probabilities of rewards and penalties (i.e., low and high penalty trials) in CG but not in AUD participants. ▪ Self-reported happiness ratings reflected the relative reward/penalty ratio in CG but not in AUD participants. ▪ In both groups (AUD and CG), self-reported sadness was minimal. Zhu et al. (2016) [47]: ▪ Money earned during task: AUD = CG ▪ Number of busts in low-penalty trials: AUD = CG ▪ Risk tolerance (time between first and second press): AUD = CG	−
	**CG** (*n* = 17) Age (y): 33.5 (SD n.r.) Sex (m/f): 10/7 Education: n.r.						
	Zhu et al. (2016) [47]: **AUD^#^** (*n* = 16) Age (y): 32.9 ± 7.2 Sex (m/f): 9/7 Education: n.r.						
	**CG** (*n* = 34) Age (y): 31.9 ± 5.7 Sex (m/f): 16/18 Education: n.r.						
Bowden-Jones et al. (2005) [77]	**AUD^#^** (*n* = 21) Age (y): 40.95 ± 9.47 Sex: n.r. Education: n.r. *Relapse at 3 months* *post-detoxification:* **AUD^#^ abstaining** (*n* = 15) Age (y): 43.87 ± 8.43 Sex: n.r. Education: n.r. **AUD^#^ relapsing** (*n* = 6) Age (y): 33.66 ± 8.40 Sex: n.r. Education: n.r.	Borderline or dissocial personality disorder: *n* = 6 Exclusion criteria for the AUD group included polysubstance dependency, an organic brain disease, learning difficulties, or a comorbid mental illness.	n.r.	Days: 21	IGT CGT	▪ IGT number of disadvantageous/risky choices: AUD = CG ▪ CGT height of bets placed across all odds: AUD = CG ▪ IGT number of disadvantageous/risky choices: AUD relapsing > AUD abstaining ▪ CGT height of bets placed across all odds: AUD relapsing > AUD abstaining ▪ CGT quality of decision-making: AUD relapsing = AUD abstaining	−
	**CG** (*n* = 20) Age (y): 36.5 ± 10.97 Sex: n.r. Education: n.r.						
Brevers et al. (2014) [70]	**AUD^#^** (*n* = 30) Age (y): 44.48 ± 11.69 Sex (m/f): 22/8 Education (y): 14.06 ± 2.63	Depression (BDI score): AUD > CG Anxiety (STAI scores): AUD > CG Exclusion criteria for the AUD group included current Axis I diagnosis, a history of severe medical illness or severe head injury.	AUD duration (y): 19.57 ± 7.17 Alcohol consumption (drinks per day): 15.13 ± 4.56 Number of times entering detoxification program: 2.31 ± 1.67	Days: 22.07 ± 3.49 (minimum: 18)	IGT CFT/LA(G)T CT	▪ IGT net score of advantageous decks: AUD < CG ▪ IGT block scores: AUD performed worse than CG on blocks 3–5. ▪ IGT increase in task performance (selection from advantageous decks) over blocks/time: AUD < CG ▪ CFT/LA(G)T height of acceptance to gamble in high(er)-risk trials: AUD > CG ▪ CFT/LA(G)T: In both the AUD group and CG, risk acceptance was significantly dependent on the potential win/loss ratio. ▪ CT risk taking: AUD > CG ▪ CT: In both the AUD and CG, risk taking was significantly dependent on the level of the expected value, and risk taking was lower in the loss domain. ▪ CT: The AUD group took significantly more risk than the CG in the risk-equal and risk-advantageous conditions of the gain domain only. Associations: ▪ In a combined group of AUD and CG participants, no significant correlations were found between the risky decision-making tasks, depression (BDI), anxiety (STAI) scores, the number of cigarettes consumed per day, age, sex, and level of education.	+
	**CG** (*n* = 30) Age (y): 41.53 ± 10.21 Sex (m/f): 24/6 Education (y): 15.10 ± 2.16						
Brière et al. (2019) [78]	**AUD*** (*n* = 67) Age (y): 50.7 ± 11.1 Sex (m/f): 51/16 Education (y): n.r.	The main psychiatric histories in the AUD and CG included mood disorders (major depressive disorder or bipolar disorder) or suicide attempts. (Co)addictions were present in both the AUD and CG.	AUD duration (y): 15.2 ± 10.7 Age of first use (y): 15.6 ± 4.4 Age of first intoxication (y): 19.1 ± 7.2 Number of previous detoxifications: 4 ± 3.9	n.r.	IGT	▪ Total net score: AUD < CG ▪ Net score block 1: AUD > CG ▪ Net score block 2: AUD = CG ▪ Net score block 3, 4, 5: AUD < CG Associations: ▪ IGT performance was found to not be dependent on possible blackout history, presence of co-addictions, currently administered psychotropic treatment, personal psychiatric history, or a family history of psychiatric problems. ▪ In the two AUD subgroups that were created based on the clinical programs that were followed (i.e., a relapse prevention program or a harm reduction program), the IGT net scores did not correlate significantly with age of first alcohol consumption, age of first intoxication, maximum previous duration of abstinence, a loss of control at 4 weeks, the duration of AUD, number of previous detoxifications, severity criteria of AUD according to the DSM-V, or the number of AUD criteria according to the DSM-V. ▪ No difference was found between men and women in IGT performance in the AUD group.	+
	**CG** (*n* = 31) Age (y): 51 ± 13.4 Sex (m/f): 15/16 Education (y): n.r.						
Burnette et al. (2021) [17]	**AUD^#^** (*n* = 16) Age (y): 31 ± 9.05 Sex (m/f): 11/5 Education (y): 15 ± 2.56	Exclusion criteria for both participant groups included systemic, neurological, cardiovascular, or pulmonary disease, or current major psychiatric/Axis I diagnoses within the last year.	Drinks per drinking day in last month: 6.45 ± 2.06	Days: 2.31 ± 1.49 (minimum: 24 h)	BART	▪ Average adjusted pumps: AUD = CG ▪ Overall amount earned: AUD = CG ▪ Number of trials presented in 10 min: AUD = CG ▪ Balloons chosen of each type/colour: AUD = CG ▪ Number of explosions: AUD = CG ▪ Risk level: Both groups (AUD and CG) more often chose for balloons with a lower explosion probability as compared to balloons with a higher explosion probability.	−
	**CG** (*n* = 16) Age (y): 30.94 ± 10.39 Sex (m/f): 11/5 Education (y): 14.31 ± 1.45						
Cordovil De Sousa Uva et al. (2010) [79]	**AUD^#^** (*n* = 35) Age (y): 48.4 ± 8.2 Sex (m/f): 17/18 Education: Secondary education: 42.85% Higher education: 57.14%	n.r.	Alcohol craving (OCDS score): AUD-T1 > AUD-T2 AUD-T1 > CG-T1, CG-T2 AUD-T2 > CG-T1, CG-T2	All variables were tested twice (i.e., at T1 and T2), except for the IGT. For analysis of the IGT, the AUD group was split into two groups, which performed the IGT either at T1 (AUD-T1) or at T2 (AUD-T2). T1—Onset of withdrawal (1–2 days of abstinence) T2—End of withdrawal (14–18 days of abstinence) AUD participants who relapsed during the program were excluded.	IGT	▪ IGT total net score: AUD-T1, AUD-T2 < CG ▪ Number of choices from disadvantageous decks: ▪ AUD-T1, AUD-T2 > CG ▪ Net scores block 1, 2, 3, 5: AUD-T1—AUD-T2 = CG ▪ Net scores block 4: AUD-T1, AUD-T2 < CG	+
	**CG** (*n* = 22) Age (y): 44.36 ± 9.64 Sex (m/f): 14/8 Education: Secondary education: 31.81% Higher education: 68.18%						
Czapla et al. (2016) [80]	**AUD^#^** (*n* = 94) Age (y): 48.05 ± 9.26 Sex (m/f): 76/18 Education (y): 12.99 ± 2.62	Exclusion criteria for both participant groups included drug abuse or dependence and severe somatic, neurological or psychiatric diseases.	AUD duration (y): 11.45 ± 10.16 AUD severity (ADS score): 15.85 ± 6.97 Drinking days in past 3 months: 51.94 ± 24.02 Cumulative amount of alcohol in past 3 months (grams): 9294.03 ± 6397 Number of prior detoxifications: 5.83 ± 7.48	Days: 18.20 ± 10.05 (range: 6–76)	CGT	▪ Quality of decision-making: AUD = CG ▪ Deliberation time: AUD > CG ▪ Risk adjustment: AUD = CG ▪ Delay aversion: AUD = CG ▪ Risk taking: AUD = CG Associations: ▪ In the six months following the first test session (i.e., follow-up) a significant decline in CGT deliberation time was observed in the AUD group. No significant interaction effect with relapse during follow-up was observed.	−
	**CG** (*n* = 71) Age (y): 46 ± 12.02 Sex (m/f): 54/17 Education (y): 13.63 ± 3.41						
Dinesh et al. (2022) [81]	**AUD* (no ADHD)** (*n* = 28) Age (y): 38.9 ± 8.2 Sex: Male (all participants) Education: School: 78.6% College: 21.4% **AUD* + ADHD** (*n* = 30) Age (y): 35.3 ± 9.0 Sex: Male (all participants) Education: School: 70% College: 30%	Exclusion criteria for the AUD groups included the presence of comorbid major psychiatric and neurological disorders, with the exception of ADHD in the AUD + ADHD group.	Duration of dependence (y): AUD total: 30.7 ± 7.8 AUD: 31.6 ± 7.9 AUD + ADHD: 29.9 ± 7.7 Duration of use (y): AUD total: 16.9 ± 8.7 AUD: 17.9 ± 9 AUD + ADHD: 16.1 ± 8.5 Age of onset AUD (y): AUD total: 20.1 ± 5.2 AUD: 21 ± 4.9 AUD + ADHD: 19.2 ± 5.5 Quantity (grams per day): AUD total: 92.5 ± 61.7 AUD: 79.7 ± 51.6 AUD + ADHD: 104.5 ± 68.5	n.r.	IGT BART	▪ IGT net scores block 1–5: AUD, AUD + ADHD < CG ▪ BART adjusted number of pumps (adaptive risk taking): AUD = AUD + ADHD = CG ▪ BART total number of balloons that popped (maladaptive risk taking): AUD = CG AUD + ADHD > CG	+/−
	**CG** (*n* = 28) Age (y): 38.5 ± 10.9 Sex: Male (all participants) Education: School: 35.7% College: 64.3%						
Dom et al. (2006) [82]	**AUD^#^ (no personality disorder)** (*n* = 38) Age (y): 43 ± 11 Sex (m/f): 74%/26% Education (y): 13.3 ± 2.5	Part of group assignment: AUD and CG: no personality disorders. AUD + A/C: AUD diagnosis and ≥1 cluster A or C personality disorder. AUD + B: AUD diagnosis and ≥1 cluster B personality disorder (borderline *n* = 18, ASP *n* = 3, both *n* = 2). Exclusion criteria for all participant groups included a current or lifetime history of psychotic disorders, amnestic disorders, neurological disorders, and severe somatic disorders.	AUD duration (y): AUD: 14.9 ± 8.4 AUD + A/C: 17.7 ± 3.1 AUD + B: 17.2 ± 9.5	Range (weeks): 4–5	IGT	▪ IGT total net score: all AUD groups < CG ▪ IGT total net score: AUD + A/C > AUD > AUD + B ▪ Over time/across blocks, CG participants showed a clear learning effect (i.e., shifted their card choices to the low-risk decks), while the AUD groups showed no learning effect over time/across blocks (significant between-group effect). The three AUD groups showed a similar (absence of a) learning effect (no significant between-group effect). Associations: ▪ The total IGT net score did not correlate significantly with age and gender in the total sample (AUD groups and CG). ▪ The total IGT net score correlated significantly with years of education in the total sample (AUD groups and CG). ▪ In the AUD groups, IGT total score did not correlate significantly with years of alcohol abuse.	+
	**AUD^#^ + A/C** (*n* = 19) Age (y): 41 ± 11 Sex (m/f): 74%/26% Education (y): 13.5 ± 3.5						
	**AUD^#^ + B** (*n* = 23) Age (y): 40 ± 7 Sex (m/f): 65%/35% Education (y): 12.5 ± 2.8						
	**CG** (*n* = 53) Age (y): 41 ± 11 Sex (m/f): 49%/51% Education (y): 13.5 ± 2.5						
Fama et al. (2016) [83]	**AUD^#^** (*n* = 39) Age (y): 48.4 ± 9.7 Sex (m/f): n.r. Education (y): 13.5 ± 2.3	Depression (BDI score): AUD > CG Lifetime criteria (not current) for nonalcohol substance abuse or dependence: 56.4% Exclusion criteria for all participant groups included a significant history of medical, psychiatric, or neurological disorders.	Age of onset AUD (median y): 20 AUD remission time (weeks): 87 (range: 0–484) Lifetime alcohol consumption (kilograms): 1257.1 ± 918.6	n.r.	CGT	▪ Quality of decision-making: AUD < CG ▪ Efficiency ratios (accuracy/response time): AUD < CG Associations: ▪ A significant negative correlation was observed between age and CGT performance in the AUD group.	+
	**CG** (*n* = 31) Age (y): 44.1 ± 9.8 Sex (m/f): n.r. Education (y): 15.1 ± 1.9						
Fein et al. (2004) [84]	**AUD^#^ Male** (*n* = 26) Age (y): 45 ± 1.34 Education (y): 15.5 ± 0.43	Exclusion criteria for all participant groups included a history or presence of an Axis I diagnosis (including ASP and CD), a significant history of head trauma or cranial surgery, a history of diabetes, stroke or hypertension, or of another neurological disease, evidence of Wernicke–Korsakoff syndrome, and current substance abuse.	Duration of use (months): AUD Male: 251 ± 17.4 AUD Female: 246 ± 21.2 Average dose (drinks/month): AUD Male: 167 ± 26.3 AUD Female: 134 ± 19.8 Duration of peak use (months): AUD Male: 52.3 ± 6.29 AUD Female: 82.8 ± 19.2 Peak dose (drinks/month): AUD Male: 353 ± 51.5 AUD Female: 311 ± 66.8	Years: AUD Male: 6.79 ± 1.19 AUD Female: 7.13 ± 1.36 Minimum (months): 6	IGT	▪ IGT total net score: AUD groups < CG ▪ IGT total net score: Female groups > Male groups Associations: ▪ A significant negative correlation was found between IGT performance and duration of peak alcohol use, and IGT performance and duration of alcohol use (the latter became insignificant after controlling for age). No significant correlation was observed between IGT performance and duration of abstinence.	+
	**AUD^#^ Female** (*n* = 18) Age (y): 47.4 ± 1.34 Education (y): 15.6 ± 0.5						
	**CG Male** (*n* = 21) Age (y): 43.1 ± 1.36 Education (y): 16 ± 0.4						
	**CG Female** (*n* = 37) Age (y): 44.8 ± 1.11 Education (y): 16.4 ± 0.28						
Fein et al. (2006) [85]	**AUD^#^** (*n* = 58) Age (y): 31.1 ± 7.8 Sex (m/f): 34/24 Education (y): 16.2 ± 1.5	Exclusion criteria for all participant groups included a lifetime or current diagnosis of schizophrenia or schizophreniform disorder, a history of drug dependence or abuse, a significant history of head trauma or cranial surgery, a history of diabetes, stroke or hypertension, or of another neurological disease, evidence of Wernicke–Korsakoff syndrome, and current substance abuse.	Age when first meeting criteria for heavy use (y): 21.2 ± 4.9 Level at first heavy use: 135.9 ± 42.4 Duration of active drinking (months): 181.2 ± 95.2 Average lifetime drinking dose (standard number of drinks/month): 84.9 ± 43.3 Duration of peak drinking (months): 55.6 ± 55.1 Peak drinking dose (standard number of drinks/month): 150.9 ± 113.1	Minimum (hours): 24	IGT	▪ IGT total net score: AUD = CG ▪ IGT total net score: Females = Males Associations: ▪ In the AUD sample, no significant correlations were observed between the IGT performance and the alcohol-use-related variables.	−
	**CG** (*n* = 58) Age (y): 31.3 ± 7.9 Sex (m/f): 34/24 Education (y): 16.5 ± 1.8						
Fishbein et al. (2007) **^a^** [48]	**AUD^#^** (*n* = 102) Age (y): 32.1 ± 5.3 Sex (m/f): 78/24 Education: Dropped high school: 16.7% College: 68.6% University: 6.9%	Psychiatric symptoms (BPRS scores): Clinical groups > CG Exclusion criteria for the clinical groups included an Axis I psychiatric disorder, severe head injury, or other sources of serious brain damage, or HIV/AIDS. Participants from the clinical groups with Axis 2 psychiatric disorders were not excluded.	AUD duration (y): AUD: 7 ± 3.8 HD + AUD: 4.3 ± 2.5 Duration of use (y): AUD: 13.8 ± 5.3 HD + AUD: 10.2 ± 4.0 Age at first use (y): AUD: 17.1 ± 2.9 HD + AUD: 16.2 ± 3.6 Number of hospitalisations for AUD: AUD: 1.95 ± 0.27 HD + AUD: 0.78 ± 0.13	Weeks: 3	CGT	▪ Percentage of high-risk choices (i.e., trials with a greater likelihood of losing than gaining points): AUD = CG HD > CG HD + AUD = CG ▪ Reaction times for high-risk choices: AUD > CG HD > CG HD + AUD = CG	−
	**HD** (*n* = 100) Age (y): 25.6 ± 4.2 Sex (m/f): 64/36 Education: Dropped high school: 31% College: 52% University: 13%						
	**HD + AUD^#^** (*n* = 60) Age (y): 26.2 ± 4.1 Sex (m/f): 53/7 Education: Dropped high school: 36.7% College: 41.6% University: 0%						
	**CG** (*n* = 160) Age (y): 28.3 ± 5.1 Sex (m/f): 122/38 Education: Dropped high school: 13.8 College: 67.5 University: 18.1						
Flannery et al. (2007) **^a^** [49]	**AUD^#^** (*n* = 102) Age (y): 32.2 ± 5.1 Sex (m/f): 78/24 Education: College or university: 77% *Subdivided by gender:* **AUD^#^ Females** (*n* = 24) Age (y): 29.65 ± 4.81 Education: College or university: 43.5% **AUD^#^ Males** (*n* = 78) Age (y): 32.97 ± 4.96 Education: College or university: 87%	Psychiatric symptoms (BPRS scores): AUD > CG Participants from the AUD group were excluded if they had an Axis I psychiatric disorder, head injury, or HIV/AIDS.	Age at first use (y): AUD: 17.5 ± 3.8 *Subdivided by gender:* AUD duration (y): AUD Females: 5.17 ± 2.74 AUD Males: 7.53 ± 3.96 Duration of use (y): AUD Females: 10.61 ± 3.9 AUD Males: 14.79 ± 5.19 Age at first use (y): AUD Females: 17.3 ± 5.24 AUD Males: 17.61 ± 3.24 Number of prior treatments for AUD: AUD Females: 0.78 ± 0.42 AUD Males: 0.69 ± 0.46 Pattern of use (binge drinkers): AUD Females: 91.3% AUD Males: 71.8%	Weeks: 3	CGT	▪ Percentage of high-risk choices (i.e., trials with a greater likelihood of losing than gaining points): AUD Females = AUD Males = CG ▪ Reaction times for high-risk choices: AUD > CG AUD Females > AUD Males AUD Females > Control Males AUD Females > Control Females	See [48] **^a^**
	**CG** (*n* = 68) Age (y): 32.9 ± 2.9 Sex (m/f): 48/20 Education: College or university: 95.6%						
Galandra et al. (2020) [56]	**AUD*** (*n* = 26) Age (y): 46.5 ± 8.25 Sex (m/f): 16/10 Education (y): 10.88 ± 3.51	Exclusion criteria for both participant groups included a presence or (family) history of neurological or (comorbid) psychiatric disorders, past brain injury or major medical disorders.	AUD duration (y): 10.77 ± 6.78 Daily alcohol dose: 15.42 ± 7.93	Days: 14.27 ± 3.91 (minimum: 10)	WoF	▪ Response times both feedback conditions: AUD > CG ▪ Number of time-outs both feedback conditions: AUD = CG ▪ Total financial outcome both feedback conditions: AUD = CG ▪ Learning curves: both participant groups showed the fastest task performance at run 2, suggesting no significant group difference in the amount of time needed to stabilize performance. Based on two choice models: ▪ Model 1: AUD and CG participants were shown to choose by maximizing expected value, but AUD participants failed to minimize disappointment and regret, whilst CG participants showed a significant anticipation of disappointment (but not regret). ▪ Model 2: AUD and CG participants were shown to choose by maximizing expected value in both feedback conditions, but in the partial feedback condition, choices of the AUD group (and not the CG) were also guided/biased by the emotional experience associated with near-miss outcomes. Contrastingly, in the complete feedback condition, the choices of the CG (but not the AUD group) were also guided/biased by near-miss outcomes, revealing group-specific modulations of choice behaviour depending on the feedback condition.	−
	**CG** (*n* = 19) Age (y): 45.11 ± 8.69 Sex (m/f): 8/11 Education (y): 10.63 ± 3.06						
Galandra et al. (2021) [50]; Canessa et al. (2021) [51] All participants from Galandra et al. (2021) [50] were analysed in Canessa et al. (2021) [51], with the addition of 1 control participant.	**AUD*** (*n* = 22) Age (y): 45.59 ± 7.99 Sex (m/f): 59%/41% Education (y): 9.91 ± 2.65	Exclusion criteria for both participant groups included major medical or neuro-psychiatric conditions, comorbid disorders with the exception of nicotine dependence, and a prior loss of consciousness or brain injury.	Duration of use (y): 10.11 ± 6.57 Daily alcohol dose: 14.34 ± 6.66 *Subdivided by gender*: Duration of use (y): AUD Females: 11.89 ± 7.11 AUD Males: 10.11 ± 7.48 Daily alcohol dose: AUD Females: 14.49 ± 5.52 AUD Males: 14.18 ± 7.12	Days: 14 ± 3.9 *Subdivided by gender:* Days: AUD Females: 14.22 ± 5.04 AUD Males: 18.92 ± 17.49	CGT	▪ Deliberation time: AUD > CG ▪ Other outcome measures (quality of decision-making, delay aversion, overall proportion bet, risk adjustment, risk taking): AUD = CG Associations: ▪ No significant correlations were observed between the deliberation times on the CGT in the AUD group and any of the alcohol consumption-related variables (duration of use, daily alcohol dose, duration of abstinence). Canessa et al. (2021) [51]: ▪ No significant correlation was observed between the deliberation times on the CGT and age in the AUD group. ▪ None of the CGT outcome measures were found to be associated with a significant group-by-sex interaction.	−
	Galandra et al. (2021) [50]: **CG** (*n* = 18) Age (y): 44.83 ± 8.86 Sex (m/f): 56%/44% Education (y): 10.11 ± 2.78 Canessa et al. (2021) [51]: CG (*n* = 19) Age (y): 45.11 ± 8.69 Sex (m/f): 50%/50% Education (y): 10.11 ± 2.78						
Genauck et al. (2017) [34]	**AUD^#^** (*n* = 15) Age (y): 45.4 ± 10.2 Sex: Male (all participants) Education (y): 16.6 ± 4.7	Depression (BDI score): AUD > CG Exclusion criteria for all participant groups included a known history of a neurological disorder or a current psychological disorder (Axis I).	AUD severity (ADS score): 19 ± 6.7 Alcohol Craving (OCDS score): 5.3 ± 4.5	Days: 42 (SD n.r.)	CFT/LA(G)T	▪ Loss aversion: AUD < CG ▪ Behavioural loss sensitivity: AUD < CG ▪ Behavioural gain sensitivity: AUD = CG ▪ Reaction times: CG participants did not change their reaction times in response to gains or losses, whilst AUD participants showed faster reaction times with increasing gains and slower reaction times with increasing losses. Associations: ▪ The AUD severity measures (ADS sum score and OCDS total scores) did not correlate significantly with loss aversion in the AUD group.	+
	**CG** (*n* = 17) Age (y): 38.8 ± 11.5 Sex: Male (all participants) Education (y): 16.3 ± 3.4						
Gilman et al. (2015) [59]	**AUD^#^** (*n* = 18) Age (y): 30.67 ± 7.10 Sex: 12/6 Education: n.r.	Mood disorders: 39% Anxiety disorders: 44% Exclusion criteria for both participant groups included psychosis, evidence of foetal alcohol spectrum disorder, chronic medical conditions, history of significant head injury or of neurological disorder.	Average number of drinks per drinking day: 13.89 ± 10.15	Minimum (days): 6 Maximum (weeks): 4	LRT	▪ Number of safe choices: AUD = CG ▪ Number of risky choices: AUD = CG ▪ Average money gained: AUD = CG ▪ Both the AUD and CG were significantly more likely to make a risky than a safe choice after a loss in the previous trial. ▪ Both the AUD and CG were equally likely to make a risky or safe choice after a win in the previous trial. ▪ No significant between-group differences were observed in the reported feelings of the participants after making a safe or risky choice. ▪ Neither the CG nor the AUD participants reported on using a specific strategy during the task.	−
	**CG** (*n* = 18) Age (y): 30.50 ± 5.06 Sex: 12/6 Education: n.r.						
Gonzalez et al. (2007) [86]	Substance-dependent group, using alcohol ≥80% of the time. **SD-AUD^#^** (*n* = 17) Age (y): 33.8 ± 2.2 Sex (m/f): 12/5 Education (y): 12.4 ± 0.51	History of previous drug abuse/dependence: 71% Lifetime prevalence mood/anxiety disorder: 93% Current major depressive disorder: 71% Exclusion criteria for both participant groups included a history of psychosis, head injury, or seizure disorder.	Age at alcohol use onset (y): 20.6 ± 1.4	Days: 26.6 ± 4.0 (minimum: 14)	IGT	▪ Total number of disadvantageous choices: SD-AUD = CG ▪ Number of disadvantageous choices block 1, 2, 3, 5: ▪ SD-AUD = CG ▪ Number of disadvantageous choices block 4: SD-AUD > CG ▪ Learning rates: CG participants shifted to and maintained selection from advantageous decks, whilst SD-AUD participants improved more slowly across decks (in the last block, the SD-AUD group made an average of 2.4 fewer choices from the disadvantageous decks as compared to the first block). Associations: ▪ Years of education was not significantly associated with IGT performance.	+/−
	**CG** (*n* = 19) Age (y): 31.1 ± 2.1 Sex (m/f): 12/7 Education (y): 14.9 ± 0.48						
Goudriaan et al. (2005) [73]	**AUD^#^** (*n* = 46) Age (y): 47.4 ± 8.4 Sex (m/f): 78%/22% Education (y): n.r.	Depression (BDI score): AUD > CG Anxiety (STAI scores): AUD > CG ADHD symptoms (ADHD-RS): AUD > CG Exclusion criteria for all participant groups included a history of major psychiatric disorders, current treatment for mental disorders, or physical conditions known to influence cognition. The AUD group did not fulfil criteria for pathological gambling or Tourette’s syndrome. Apart from an AUD group, the study included two separate clinical groups for these two disorders.	n.r.	Minimum (months): 3 Maximum (months): 12	IGT CPT	▪ IGT total number of choices from advantageous decks: AUD < CG ▪ IGT learning rates: Both groups (AUD and CG) learned to choose from advantageous decks across blocks. ▪ IGT: Both groups (AUD and CG) more often chose from the infrequent punishment decks than from the frequent punishment decks. ▪ IGT conceptual knowledge (i.e., correct identification of (dis)advantageous decks at end of task): AUD < CG ▪ IGT response times: AUD = CG ▪ IGT response times after rewards and net losses did not differ between groups; both groups (AUD and CG) showed higher percentages of changing decks after losses than after rewards. ▪ IGT response times after rewards were faster than after losses in both groups (AUD and CG). ▪ IGT self-reported motivation to perform, task appraisal, and opinion about own performance compared to others: AUD = CG ▪ CPT performance was significantly worse in the AUD group than in the CG because more AUD participants used a conservative card selection strategy. ▪ CPT response times: while the CG slowed down in response time after a loss as compared to after a win, the AUD group did not. ▪ CPT self-reported motivation to perform: AUD < CG ▪ CPT self-reported task appraisal after performance: AUD < CG ▪ CPT self-evaluation of how well they performed compared to others: AUD < CG Associations: ▪ Anxiety levels (STAI scores), depression severity levels (BDI scores), and ADHD symptoms (ADHD-RS scores) were not significantly correlated with IGT or CPT performance.	+
	**CG** (*n* = 49) Age (y): 35.8 ± 11.1 Sex (m/f): 69%/31% Education (y): n.r.						
Harvanko et al. (2012) [87]	**AUD^#^** (*n* = 25) Age (y): 21.16 ± 3.31 Sex (m/f): 20/25 Education: High school or below: 4% Some college: 80% College graduate or higher: 16%	Exclusion criteria for all participant groups included the presence of current Axis-I disorders.	Frequency of alcohol use (times per week): AUD: 1.90 ± 1.65 At-Risk Drinkers: 1.79 ± 0.92	n.r.	CGT	▪ Overall proportion of bets: AUD, At-Risk Drinkers > CG ▪ Quality of decision-making: AUD = At-Risk drinkers = CG	+/−
	**At-Risk Drinkers**^6^ (*n* = 82) Age (y): 22.12 ± 2.99 Sex (m/f): 62/20 Education: High school or below: 3.66% Some college: 59.76% College graduate or higher: 36.59%						
	**CG** (*n* = 48) Age (y): 19.50 ± 2.43 Sex (m/f): 33/15 Education: High school or below: 8.33% Some college: 85.42%% College graduate or higher: 4.17%						
Kamarajan et al. (2010) [57]	**AUD^#^** (*n* = 40) Age (y): 38.28 ± 6.44 Sex: Male (all participants) Education (y): n.r.	Exclusion criteria for both participant groups included a personal and/or family history of major medical or psychiatric disorders and substance-related addictive illnesses.	n.r.	Minimum (days): 28	SOG	▪ Selection frequencies for ‘10’ and ‘50’ after single or consecutive loss/gain trials: AUD = CG ▪ Reaction times: AUD = CG	−
	**CG** (*n* = 40) Age (y): 21.07 ± 3.36 Sex: Male (all participants) Education (y): n.r.						
Khemiri et al. (2020) [88]	**AUD^#^** (*n* = 106) Age (y): 47.9 ± 7.5 Sex (m/f): 51.9%/48.1% Education: Did not finish primary school: 0% Primary school: 17% Secondary school: 34% University: 49.1%	Depressive symptoms (MADRS score): AUD > CG Exclusion criteria for both groups included a diagnosis of substance abuse or dependence, severe major psychiatric disorder, severe somatic illness, history of stroke, intracranial haemorrhage, or severe head trauma.	Age of onset alcohol problem: 34.1 ± 10.6 Alcohol craving (OCDS scores): 23.8 ± 6.5 Years with current level of drinking: 7.7 ± 6.6 Drinking days in last 90 days: 72% ± 21 Heavy drinking days in last 90 days: 62% ± 27	Range (days): 4–14	CGT	▪ Deliberation time: AUD > CG ▪ Risk-taking AUD = CG ▪ Overall proportion bet: AUD = CG	−
	**CG** (*n* = 90) Age (y): 48.1 ± 11.8 Sex (m/f): 43.3%/56.7% Education: Did not finish primary school: 1.1% Primary school: 4.5% Secondary school: 46.1% University: 48.3%						
Kim et al. (2006) [64]	**AUD^#^** (*n* = 28) Age (y): 40.8 ± 5.6 Sex: Male (all participants) Education (y): 12.7 ± 1.5	ASPD: AUD + CD: *n* = 7 Exclusion criteria for all participant groups included past neurological illness or traumatic brain injury, and lifetime Axis I psychiatric diagnoses.	AUD severity (MAST score): AUD: 27.4 ± 13.4 AUD + CD: 27.1 ± 8.7 Alcohol Craving (OCDS score): AUD: 23.3 ± 7.3 AUD + CD: 24.9 ± 10.2 Age at first drinking (y): AUD: 19.6 ± 1.8 AUD + CD: 17.8 ± 1.9 Age of onset problematic drinking (y): AUD: 32.6 ± 7.1 AUD + CD: 31.2 ± 7.5 Duration of problematic drinking (y): AUD: 8.1 ± 5.6 AUD + CD: 7.6 ± 5.9 Age at first detoxification (y): AUD: 37.9 ± 6.2 AUD + CD: 35.8 ± 6.8 Number of prior detoxifications: AUD: 4.5 ± 5.4 AUD + CD: 4.4 ± 5.6	Minimum: 2 weeks	IGT IGT variant	▪ IGT total net score: AUD, AUD + CD, CG + CD < CG ▪ IGT net scores block 1, 2: AUD = AUD + CD = CG + CD = CG ▪ IGT net scores block 3: AUD, CG > AUD + CD, CG + CD ▪ IGT net scores block 4, 5: AUD, AUD + CD, CG + CD < CG ▪ IGT learning rates across blocks: AUD, AUD + CD, CG + CD < CG ▪ As compared to the CG, the AUD group showed a delayed choice shift to the advantageous card decks, whilst the AUD + CD and CG + CD groups showed minimal learning across blocks. ▪ IGT variant total net score: AUD = AUD + CD = CG + CD = CG ▪ IGT variant net scores block 1–5: AUD = AUD + CD = CG + CD = CG Associations: ▪ The alcohol-use-related questionnaires (MAST, OCDS) did not correlate significantly with IGT or IGT variant performance (net scores). ▪ None of the alcohol-use-related variables significantly predicted IGT or IGT variant performance (net scores).	+/−
	**AUD^#^ + CD**^7^ (*n* = 28) Age (y): 38.7 ± 7.5 Sex: Male (all participants) Education (y): 10.8 ± 1.4						
	**CG + CD**^7^ (*n* = 10) Age (y): 42.3 ± 7.3 Sex: Male (all participants) Education (y): 12.8 ± 1.3						
	**CG** (*n* = 30) Age (y): 39.1 ± 7.3 Sex: Male (all participants) Education (y): 14.0 ± 1.8						
Kim et al. (2011) [89]	**AUD^#^** (*n* = 23) Age (y): 32.65 ± 5.10 Sex: Male (all participants) Education (y): 11.26 ± 2.77	Exclusion criteria for both participant groups included past neurological illness, traumatic brain injury, and lifetime axis I psychiatric disorders.	AUD duration (y): 4.91 ± 3.64 AUD severity (MAST score): 25.78 ± 10.16	Minimum: 2 weeks	IGT GDT	▪ IGT total net score: AUD < CG ▪ IGT selection from disadvantageous/risky decks (A and B): AUD > CG ▪ IGT selection from advantageous deck C: AUD = CG ▪ IGT selection from advantageous deck D: AUD < CG ▪ IGT net score blocks 1–2: AUD = CG ▪ IGT net score blocks 3–5: AUD < CG ▪ IGT learning rate: the performance of the CG improved across blocks, whilst the AUD group showed an impaired performance in the later blocks. ▪ GDT total number of risky choices: AUD > CG ▪ GDT number of one number choices (risky option): AUD > CG ▪ GDT number of two number choices (risky option): AUD = CG ▪ GDT number of three number choices (safe option): AUD < CG ▪ GDT number of four number choices (safe option): AUD = CG Associations: ▪ Education level correlated significantly (positive correlation) with total IGT net score (unclear whether this concerned the AUD and the CG separately or a combined group of AUD and CG participants). ▪ In the AUD group, alcohol severity (MAST total scores) did not correlate significantly with any of the IGT outcome measures.	+
	**CG** (*n* = 21) Age (y): 30.52 ± 2.98 Sex: Male (all participants) Education (y): 15.14 ± 1.20						
Körner et al. (2015) [90]	**AUD^#^** (*n* = 40) Age (y): 48.15 ± 10.52 Sex (m/f): 27/13 Education (y): 15.15 ± 2.76	The exclusion criteria for both participant groups included severe mental disorders such as psychosis, head injuries, organic brain syndrome, other cognitive disorders, and a history of seizures.	n.r.	Months: 38.45 ± 87.27 Range: 2 weeks-38 years	IGT	▪ Total financial outcome: AUD < CG ▪ Net scores block 1–5: AUD < CG ▪ Learning profile across blocks: AUD = CG Associations: ▪ IGT performance was not significantly associated with sex in AUD and CG. ▪ IGT performance was not significantly associated with duration of abstinence in two AUD subgroups (short-term (*n* = 20) and long-term (*n* = 20) abstainers). ▪ When MWTB, d2, BDI-II, STAXI, and BDHI scores were entered into one model, only the BDHI negativism subscale explained a significant proportion of variance of IGT performance in the AUD group, and none of the entered variables could significantly explain variance in IGT performance in the CG.	+
	**CG** (*n* = 40) Age (y): 45.40 ± 10.73 Sex (m/f): 27/13 Education (y): 15 ± 2.7						
Kornreich et al. (2013) [60]	**AUD^#^** (*n* = 25) Age (y): 45.68 ± 7.97 Sex (m/f): 15/10 Education: Completed 3 years of secondary school: 40% Completed secondary school: 24% Post-secondary school education: 36%	Depression (BDI score): AUD > CG Anxiety (STAI scores): AUD > CG The exclusion criteria for all participant groups included a history of bipolar disorder or schizophrenia.	Duration of alcohol use (months): 94.32 ± 84.28 Number of hospitalizations for alcohol use: 2.48 ± 2.6	Assessments took place in the third week of abstinence.	IGT	▪ Total number of advantageous choices: AUD = CG ▪ Learning rate: A significant difference in the learning rate across blocks was observed between the CG and AUD groups, specifically indicating that the CG improved task performance in block 4 and 5 of the IGT, whilst the AUD group did not perform better during the later stages of the task.	+/−
	**CG** (*n* = 25) Age (y): 38.96 ± 10.47 Sex (m/f): 19/6 Education: Completed 3 years of secondary school: 16% Completed secondary school: 32% Post-secondary school education: 52%						
Lawrence et al. (2009) [91]	**AUD^#^** (*n* = 21) Age (y): 44.2 ± 9.2 Sex: Male (all participants) Education (y): 11.9 ± 3.4	Depression (BDI score): AUD > CG Exclusion criteria for all participant groups included comorbid psychiatric illness (with the exception of depression in the AUD group), history of head injury, or a neurological disorder.	AUD severity (SADQ score): 33.7 ± 16	Self-reported duration (days): 150 (SD n.r.) Minimum: > 1 week (although 4 participants consumed alcohol in the past 48 h)	CGT	▪ Number of bankruptcies: AUD = CG ▪ Total points obtained: AUD = CG ▪ Quality of decision-making: AUD = CG ▪ Both groups (AUD and CG) were more likely to choose the colour in the majority at higher ratios. ▪ Deliberation time: AUD > CG ▪ Both groups (AUD and CG) showed a longer deliberation time when the ratio was less certain. ▪ Wagering in ascend condition: AUD = CG ▪ Wagering in descend condition: the AUD group placed significantly higher wagers than the CG, particularly in trials with unfavourable odds.	+/−
	**CG** (*n* = 21) Age (y): 40.2 ± 13.6 Sex: Male (all participants) Education (y): 13.5 ± 2.4						
Le Berre et al. (2012) [92]	**AUD^#^** (*n* = 31) Age (y): 43.87 ± 6.97 Sex (m//f): 26/5 Education (y): 10.77 ± 2.14	Depression (BDI score): AUD > CG Anxiety (STAI scores): AUD > CG Exclusion criteria for both participant groups included the display of psychiatric problems or any history of head injury, coma, epilepsy, Wernicke’s encephalopathy, cirrhosis, or depression.	AUD duration (y): 8.26 ± 8.26 Quantity (units per day): 21.95 ± 11.70	Days: 12.64 ± 7.16 (range: 7–40)	IGT	▪ Task performance: AUD < CG	+
	**CG** (*n* = 37) Age (y): 45.49 ± 6.07 Sex (m//f): 25/12 Education (y): 12.11 ± 3.56						
Le Berre et al. (2014) [93]	**AUD^#^** (*n* = 30) Age (y): 43.67 ± 7.04 Sex (m//f): 26/4 Education (y): 10.53 ± 2.32	Depression (BDI score): 8 ± 3.36 Anxiety (STAI scores): State anxiety: 33.13 ± 10.06 Trait anxiety: 50.27 ± 14.22 Exclusion criteria for both participant groups included the display of psychiatric problems or any history of head injury, coma, epilepsy, Wernicke’s encephalopathy, cirrhosis, or depression.	AUD duration (y): 9.03 ± 9.37 Duration of alcohol use (y): 27.97 ± 8.23 Duration of alcohol misuse (y): 15.60 ± 9.69 Quantity (units per day): 24.16 ± 15.44 Number of withdrawals: 2.20 ± 1.16	Days: 12.63 ± 7.08 (range: 7–40)	IGT	▪ Total net score: AUD < CG ▪ Net scores block 1–4: AUD = CG ▪ Net scores block 5: AUD < CG ▪ Learning rates: The CG showed an improvement across blocks, whilst the AUD group showed a significantly impaired performance in the 5th block. ▪ As compared to the CG, the AUD group more often selected cards from the disadvantageous/risky decks and less often from the advantageous decks. Associations: ▪ None of the alcohol-use-related measures correlated significantly with IGT performance in the AUD group.	+
	**CG** (*n* = 45) Age (y): 44.76 ± 7.78 Sex (m//f): 38/7 Education (y): 10.69 ± 2.25						
Loeber et al. (2009) **^b^** [52]	**AUD^#^** (*n* = 48) Age (y): 46.5 ± 8.2 Sex (m/f): 27/21 Education: n.r. *Subdivided in high number (≥2) of detoxifications (AUD-HD) and low number (<2) of detoxifications (AUD-LD) groups:* **AUD^#^-HD** (*n* = 27) Age (y): 47.8± 8.8 Sex (m/f): 19/8 Education: n.r. **AUD^#^-LD** (*n* = 21) Age (y): 45 ± 7.2 Sex (m/f): 8/13 Education: n.r. *Subdivided in recently (≤16 days) abstinent (AUD-RA) and longer (>16) abstinent (AUD-LA) groups:* **AUD^#^-RA** (*n* = 28) Age (y): 45.3 ± 7.3 Sex (m/f): 15/13 Education: n.r **AUD^#^-LA** (*n* = 20) Age (y): 48.3 ± 9.2 Sex (m/f): 12/8 Education: n.r	Depression (BDI score): AUD > CG AUD-HD, AUD-LD, AUD-RA, AUD-LA > CG Exclusion criteria for both participant groups included current drug abuse or dependence, severe somatic, neurological or psychiatric diseases, or suicidal tendencies.	AUD duration (y): AUD: n.r. AUD-HD: 17.4 ± 9.4 AUD-LD: 9.4 ± 6 AUD-RA: 13 ± 9.2 AUD-LA: 15.2 ± 8.7 AUD severity (ADS total score): AUD: 15.2 ± 6.8 AUD-HD: 17 ± 7.1 AUD-LD: 12.9 ± 5.8 AUD-RA: 16.2 ± 7.3 AUD-LA: 13.8 ± 6 Age of onset regular alcohol use: AUD: 19.4 ± 4.5 AUD-HD: 19.6 ± 5.2 AUD-LD: 19.1 ± 3.3 AUD-RA: 19.3 ± 4.4 AUD-LA: 19.6 ± 4.6 Lifetime alcohol consumption (number of drinks): AUD: 91,987.6 ± 102,845.9 AUD-HD: 98,694.3 ± 85,205.3 AUD-LD: 83,364.6 ± 123,611.9 AUD-RA: 93,313.9 ± 117,925.1 AUD-LA: 90,130.6 ± 79,981.5 Number of prior detoxifications: AUD: n.r. AUD-HD: 7.1 ± 11 AUD-LD: 0.2 ± 0.4 AUD-RA: 4.4 ± 10.7 AUD-LA: 3.6 ± 5.5	Days: AUD: 15.65 ± 6.69 AUD-HD: 16.2 ± 8 AUD-LD: 14.9 ± 4.6 AUD-RA: 11.5 ± 3.8 AUD-LA: 21.4 ± 5.5	IGT	▪ Net scores block 1–5: AUD = CG ▪ Net scores block 1–5: AUD-HD = AUD-LD ▪ Total net score: AUD-RA < AUD-LA, CG ▪ Learning rate across blocks: AUD = CG ▪ Learning rates: in both groups (AUD and CG), performance increased across blocks and significant differences were observed between block 1 and 2, but not between block 2 and 3, 3 and 4, or 4 and 5. ▪ Learning rates across blocks: AUD-HD < AUD-LD ▪ Learning rates: both the AUD-HD and AUD-LD improved performance across blocks, and both groups performed equally well in the last 10 trials. ▪ Learning rates: AUD-RA = AUD-LA = CG (all three groups improved across blocks). Associations: ▪ Depression (BDI total scores) was not significantly correlated with IGT performances in the AUD group and CG. ▪ Severity of nicotine dependence was not significantly correlated with IGT performances in the AUD group and CG.	−
	**CG** (*n* = 36) Age (y): 44.4 ± 9.9 Sex (m//f): 23/13 Education: n.r.						
Loeber et al. (2010) **^b^** [53]	**AUD^#^-HD** (*n* = 31) Age (y): 47.4 ± 8.4 Sex (m/f): 76%/24% Education: n.r.	Depression (BDI score): AUD-HD > CG AUD-LD = CG Exclusion criteria for all participant groups included current drug abuse or dependence, severe somatic, neurological or psychiatric diseases, or suicidal tendencies.	AUD severity (ADS total score): AUD-HD: 16.7 ± 6.8 AUD-LD: 12.6 ± 6.1 Age of onset regular alcohol use: AUD-HD: 19.4 ± 5 AUD-LD: 19.4 ± 3.5 Lifetime alcohol consumption (number of drinks): AUD-HD: 94,065.5 ± 80,793.8 AUD-LD: 88,198.4 ± 137,141.7 Daily dose in last 90 days prior to admission (number of alcohol units): AUD-HD: 160.4 ± 126 AUD-LD: 107.4 ± 114.5 Drinking days in last 90 days prior to admission: AUD-HD: 51.7 ± 20.2 AUD-LD: 50.7 ± 26.5 Number of prior detoxifications: AUD-HD: 7.3 ± 10.4 AUD-LD: 1 ± 0	n.r.	IGT	▪ Net score at T1 (during inpatient treatment): AUD-HD, AUD-LD = CG ▪ Net score at T2 (3 months after discharge): AUD-HD, AUD-LD = CG ▪ None of the groups (AUD groups and CG) showed significant changes in their IGT performance from T1 to T2. ▪ IGT net scores at T3 (6 months after discharge): ▪ AUD-HD = AUD-LD ▪ The AUD-HD and AUD-LD groups did not show significant changes in their IGT performance from T2 to T3. Associations: ▪ IGT net score was a significant predictor for relapse in the AUD groups, where participants with a higher net score at first assessment (T1) were more likely to relapse within the first six months after discharge.	See [52] **^b^**
	**AUD^#^-LD** (*n* = 17) Age (y): 44.9 ± 7.7 Sex (m/f): 50%/50% Education: n.r.						
	**CG** (*n* = 36) Age (y): 44.4 ± 9.1 Sex (m/f): 56.3%/43.7% Education: n.r.						
Maurage et al. (2018) [94]	**Severe AUD*** (*n* = 38) Age (y): 46.95 ± 11.30 Sex (m/f): 29/9 Education (y): 12.50 ± 2.42 *Subdivided by relapse status 6 months after start of detoxification:* **AUD*-Abstaining** (*n* = 17) Age (y): 43.18 ± 9.12 Sex (m/f): 12/5 Education (y): 12.35 ± 2.09 **AUD*-Relapsing** (*n* = 21) Age (y): 50 ± 12.17 Sex (m/f): 17/4 Education (y): 12.62 ± 2.71	Depressive and anxiety symptoms (HADS scores): Depression: 5.92 ± 2.07 Anxiety: 7.87 ± 3.03 Only one AUD participant presented with severe anxious symptomatology according to the HADS. AUD participants were free of any current psychiatric diagnosis.	AUD duration (y): Severe AUD: 24.68 ± 12.79 AUD-Abstaining: 21.88 ± 9.72 AUD-Relapsing: 26.95 ± 14.66 AUD severity (MAST score): Severe AUD: 31.92 ± 9.54 AUD-Abstaining: 33.12 ± 9.71 AUD-Relapsing: 30.95 ± 9.54 Dangerousness of alcohol consumption (AUDIT score): Severe AUD: 27.66 ± 7.40 AUD-Abstaining: 27.29 ± 6.56 AUD-Relapsing: 27.95 ± 8.17 Daily consumption before detoxification (number of drinks): Severe AUD: 16.45 ± 8.66 AUD-Abstaining: 15.47 ± 8.59 AUD-Relapsing: 17.24 ± 8.85 Number of prior detoxifications: Severe AUD: 1.89 ± 2.17 AUD-Abstaining: 1.29 ± 1.36 AUD-Relapsing: 2.38 ± 2.57	Days: Severe AUD: 31.66 ± 16.33 AUD-Relapsing: 34.57 ± 17.94 AUD-Abstaining: 28.06 ± 13.76 Minimum (days): 14	IGT	▪ Total net score: Severe AUD < CG ▪ Net scores block 1–2: Severe AUD = CG ▪ Net scores block 3–5: Severe AUD < CG ▪ Learning rates: The performance of the CG improved across blocks, whilst the severe AUD showed impaired performance in the last three blocks (when decision-making under uncertainty shifts to decision-making under risk). ▪ Total net score: AUD-Relapsing = AUD-Abstaining ▪ Net scores block 1–5: AUD-Relapsing = AUD-Abstaining ▪ Learning rates: AUD-Relapsing = AUD-Abstaining Associations: ▪ In the severe AUD group, IGT performance did not correlate significantly with demographics (i.e., age, gender, education), alcohol-use-related, psychopathological (HADS-scores), or physiological measures (liver stiffness scores).	+
	**CG** (*n* = 38) Age (y): 46.66 ± 13.42 Sex (m/f): 29/9 Education (y): 12.45 ± 2.88						
Miranda et al. (2009) [95]	**AUD^#^** (*n* = 22) Age (y): 29.3 ± 5.6 Sex: Male (all participants) Education (y): 12.9 ± 2.4	Part of group assignment. AUD groups: Avoidant personality disorder: 10.3% Obsessive compulsive personality disorder: 15.4% Paranoid personality disorder: 2.6% Schizoid personality disorder: 2.6% Narcissistic personality disorder: 2.6% Exclusion criteria for all participant groups included current or lifetime bipolar I or II disorder, agoraphobia, a psychotic disorder, posttraumatic stress disorder, panic disorder, obsessive compulsive disorder, and eating disorders as well as a current mood disorder, generalized anxiety disorder, or an active substance use disorder.	n.r.	Months: AUD: 13.64 ± 21.34 AUD + ASPD: 13.56 ± 21.65 Minimum (days): 30	IGT	▪ Total net scores: AUD, AUD + ASPD < CG ▪ Net scores block 1–2, 4: AUD = AUD + ASPD = CG ▪ Net scores block 3: AUD < AUD + ASPD, CG ▪ Net scores block 5: AUD + ASPD < AUD, CG ▪ Strategy insight (i.e., knowledge of which decks were disadvantageous/advantageous at end of task): AUD = AUD + ASPD = CG	+
	**AUD^#^ + ASPD** (*n* = 17) Age (y): 27.1 ± 6.1 Sex: Male (all participants) Education (y): 11.3 ± 1.4						
	**CG** (*n =* 21) Age (y): 22.6 ± 4.3 Sex: Male (all participants) Education: 14.5 ± 1.4						
Murray et al. (2018) [96]	**AUD^#^ + Smoking** (*n* = 20) Age (y): 45.9 ± 9.5 Sex (m/f): 16/4 Education: 13.8 ± 1.4	Depression (BDI score): AUD + Smoking > CG + Smoking Anxiety (STAI scores): AUD + Smoking > CG + Smoking Exclusion criteria for all participant groups included neurological disorders, psychiatric disorders, and vascular risk factors.	Duration of alcohol use (y): 29 ± 9 Lifetime average alcohol use (drinks/month): 229 ± 151 One-year average alcohol use (drinks/month): 299 ± 159	Days: 14 ± 10 (range: 1–31)	IGT BART	▪ IGT total net score: AUD + Smoking = CG + Smoking = CG Non-smoking ▪ BART task performance: AUD + Smoking = CG + Smoking = CG Non-smoking	−
	**CG + Smoking** (*n* = 35) Age (y): 47.4 ± 10.2 Sex (m/f): 31/4 Education: 14.7 ± 2.2						
	**CG Non-smoking** (*n* = 29) Age (y): 47.2 ± 11.2 Sex (m/f): 24/5 Education: 17.1 ± 2						
Noël et al. (2007) [54]; Noël et al. (2011) [55]	**AUD^#^** (*n* = 30) Age (y): 45.8 ± 9.5 Sex (m/f): 20/10 Education: 10.7 ± 2	Depression (MADRS score): AUD > CG Anxiety (STAI scores): AUD > CG Exclusion criteria for the AUD group included other current Axis I diagnoses, a history of significant medical illness, head injury, and overt cognitive dysfunction.	Years of heavy drinking: 14.05 ± 8.7/17.05 ± 8.7 ^8^ Daily quantity (grams): 352.5 ± 240.5 Number of prior detoxifications: 4.5 ± 1.7	Days: 19.3 ± 2.5 (minimum: 15) (range: 18–21)	IGT	Noël et al. (2007) [54]: ▪ Total net score: AUD < CG ▪ Total net score < 10 (impaired task performance): AUD: 47% CG: 13% ▪ Net scores block 1–4: AUD = CG ▪ Net scores block 5: AUD < CG ▪ Task performance part 1 (block 1–2): AUD = CG ▪ Task performance part 2 (block 4–5): AUD < CG ▪ Learning rates: The CG improved across blocks, whilst the AUD group did improve across blocks, but performed particularly poorly in block 5. Associations: ▪ The AUD group that was impaired on the IGT (total net score < 10; *n* = 14) had a higher daily alcohol use, a higher number of prior detoxifications, and a longer duration of AUD as compared to the unimpaired AUD group (*n =* 16), suggesting IGT performance to be associated with the severity of AUD. Noël et al. (2011) [55]: ▪ Total net score: AUD < CG ▪ Learning rates: Performance of both groups (AUD and CG) generally improved from block 1–4, but the CG performed significantly better than the AUD group on the fifth block (impaired performance in the AUD group).	+
	**CG** (*n* = 30) Age (y): 44.1 ± 8.9 Sex (m/f): 19/11 Education: 10.8 ± 2.5						
Roopesh et al. (2017) [58]	**AUD^#^** (*n* = 26) Age (y): 33.88 ± 8.28 Sex: Male (all participants) Education (y): 9.61 ± 4.11	Exclusion criteria for the AUD group included comorbid psychiatric, major medical, neurological, or neurosurgical disorders.	AUD severity (SADQ score): 31.73 ± 10.66	Days: 14 (SD n.r.)	EGT	▪ Net value of final amount gained: AUD < CG ▪ Number of choices for decks A-D: AUD = CG Associations: ▪ In the AUD group, EGT final amount gained correlated significantly with age (positive correlation). No significant correlations were observed between EGT final amount gained and education, or AUD severity (SADQ). ▪ In the AUD group, the number of choices for each of the separate decks (A-D) did not correlate significantly with age, education, or AUD severity (SADQ). ▪ In the total sample (AUD and CG), no significant correlations were observed between any of the EGT outcome measures and age or education.	+
	**CG** (*n* = 26) Age (y): 32.69 ± 6.23 Sex: Male (all participants) Education (y): 11.92 ± 4.7						
Salgado et al. (2009) [97]	**AUD^#^** (*n* = 31) Age (y): 48.97 ± 6.1 Sex (m/f): 26/5 Education (y): 10.55 ± 2.6	Exclusion criteria for both participant groups included substance-related disorders disorders, current major depressive disorder or manic/hippomanic episode, a history of psychotic disorder, obsessive-compulsive disorder, impulse control related disorders such as pathological gambling, borderline personality, attention-deficit/hyperactivity disorder or eating disorders, or a lifetime history of traumatic brain injury/vascular brain disorder.	AUD duration (y): nearly 10–15 (SD n.r.) Duration of alcohol use (y): nearly 30 (SD n.r.). Quantity of use: Men (units per week): > 51 Women (units per week): >31	Range (days): 15–120 Shorter abstinence (15–30 days): *n* = 20 Longer abstinence (60–120 days): *n* = 9	IGT	▪ Total net score: AUD < CG ▪ Net scores block 1: AUD = CG ▪ Net scores block 2: AUD < CG (less advantageous choices) ▪ Net scores block 3: AUD = CG ▪ Net scores block 4–5: AUD < CG (less advantageous choices) Associations: ▪ No significant correlation was observed between IGT performance and abstinence period.	+
	**CG** (*n* = 30) Age (y): 46.93 ± 8.3 Sex (m/f): 20/10 Education (y): 11.07 ± 4.0						
Sehrig et al. (2019) [98]	**AUD*** (*n* = 39) Age (y): 43.2 ± 8.4 Sex (m/f): 72%/28% Education (y): 10.2 ± 1.6	n.r.	n.r.	AUD participants were abstinent at time of assessment (duration: n.r.)	BART	▪ Average number of pumps: AUD = CG ▪ Number of popped balloons: AUD = CG ▪ Reaction times cash-out decisions (i.e., decision to collect money): AUD > CG ▪ Reaction times low-risk decisions: AUD > CG ▪ Reaction times high-risk decisions: AUD > CG ▪ Risk-level effects: responses of both groups (AUD and CG) were fastest when making cash-out decisions and slowest when making high-risk decisions.	−
	**CG** (*n* = 35) Age (y): 42.7 ± 9.9 Sex (m/f): 66%/34% Education (y): 12.2 ± 1.4						
Sönmez et al. (2023) [99]	**AUD*** (*n* = 85) Age (y): 46.33 ± 10 Sex: Male (all participants) Education (y): 9.35 ± 3.20	Exclusion criteria for the AUD group included having drug-related cognitive deficiencies, severe comorbid psychiatric disorders, or other medical conditions that could hamper the understanding of study instructions.	n.r.	Minimum (weeks): 3	IGT	▪ Total net score: AUD < CG Associations: ▪ In the AUD group, IGT net scores correlated significantly with alcohol craving scores on the PACS (negative correlation), and did not correlate significantly with age, education, smoking amount or history, age of onset of alcohol use, duration of regular alcohol use, or with depression or anxiety sub-scores on the API.	+
	**CG** (*n* = 87) Age (y): 45.40 ± 12.16 Sex: Male (all participants) Education (y): 9.21 ± 3.64						
Sübay et al. (2021) [100]	**AUD*** (*n* = 40) Age (y): 46.58 ± 10.06 Sex: Male (all participants) Education (y): 8.73 ± 2.99	Exclusion criteria for the AUD group included having drug-related cognitive deficiencies, severe comorbid psychiatric disorders, or other medical conditions that could hamper the understanding of study instructions.	Addiction severity (API score): 11.19 ± 2.88 Alcohol craving (PACS score): 6.88 ± 7.5 Age at onset alcohol use (y): 19.13 ± 3.98 Duration of regular alcohol use (y): 22.30 ± 10.48	Minimum (weeks): 3	IGT	▪ Total net score: AUD < CG	+
	**CG** (*n* = 40) Age (y): 37.45 ± 12.16 Sex: Male (all participants) Education (y): 9.25 ± 2.33						
Tomassini et al. (2012) [101]	**AUD^#^** (*n* = 26) Age (y): 46.15 ± 7.67 Sex (m/f): 21/6 Education (y): 9.38 ± 2.77	n.r.	Age of onset AUD (y): 23.19 ± 9.04	Months: 16.85 ± 13.21 (minimum: 6)	IGT	▪ Total net score: AUD < CG ▪ Net score block 1: AUD = CG ▪ Net score block 2: AUD < CG ▪ Net score block 3: AUD = CG ▪ Net score block 4–5: AUD < CG ▪ Learning rates: Both groups (AUD and CG) improved over the blocks, but the CG improved more than the AUD group. Associations: ▪ Educational level was not significantly correlated with IGT performance in the two groups (AUD and CG). ▪ In the AUD group, no significant correlations were observed between the clinical variables (e.g., abstinence duration) and IGT scores.	+
	**CG** (*n* = 24) Age (y): 40.08 ± 12.79 Sex (m/f): 13/11 Education (y): 12.37 ± 3.40						
Van der Plas et al. (2009) [61]	Substance-dependent group, using alcohol ≥80% of the time. **SD-AUD^#^** (*n* = 33) Age (y): 37.9 ± 9.9 Sex (m/f): 15/18 Education (y): 13.8 ± 2.4	Exclusion criteria for the SD-AUD group included presence of psychosis, past head injury or seizure disorder. The comorbidities that were assessed were psychoses, major depressive disorder, and anxiety disorders. A sum score of >3 on the SCID for each of these comorbidities led to exclusion.	AUD duration (y): 19.5 ± 10.4 Frequency of use in 30 days before assessment: 2.7 ± 3.5	Minimum (days): 15	IGT	▪ Number of advantageous choices block 1–3: SD-AUD = CG ▪ Number of advantageous choices block 4–5: SD-AUD < CG ▪ Learning rates: SD-AUD = CG Associations: ▪ No significant sex differences in IGT performance were found for the SD-AUD group and CG.	+
	**CG** (*n* = 36) Age (y): 28.9 ± 9.8 Sex (m/f): 19/17 Education (y): 16.2 ± 2.4						
Vitoria-Estruch et al. (2018) [102]	Interpersonal violence perpetrators with AUD **(AUD* + IPV)** (*n* = 28) Age (y): 40.21 ± 11.90 Sex: Male (all participants) Education; Primary/lower secondary: 71.43% Upper secondary/vocational training: 25% University: 3.57%	Exclusion criteria for the AUD + IPV group included having any mental illness and psychopathological signs. History of TBI: 48.14%	Age at first use (y): 16.25 ± 2.17 Duration of use (y): 21.83 ± 10.78 Quantity: 69.10 ± 85.60	Months: 0.33 ± 0.78	CGT	▪ Proportion of bets: AUD + IPV > CG, CG + IPV ▪ Risk-taking: AUD + IPV > CG, CG + IPV ▪ Deliberation time: AUD + IPV = CG + IPV = CG ▪ Quality of decision-making: AUD + IPV = CG + IPV = CG	+/−
	Interpersonal violence perpetrators without AUD **(CG + IPV)** (*n* = 35) Age (y): 39.34 ± 9.83 Sex: Male (all participants) Education: Primary/lower secondary: 45.71% Upper secondary/vocational training: 48.57% University: 5.72%						
	**CG** (*n* = 37) Age (y): 41.75 ± 11 Sex: Male (all participants) Education: Primary/lower secondary: 43.24% Upper secondary/vocational training: 45.95% University: 10.81%						
Xie et al. (2018) [103]	**AUD^#^** (*n* = 48) Age (y): 48.21 ± 5.89 Sex (m/f): 40/8 Education (y): 10.46 ± 1.79	Exclusion criteria for both participant groups included current or past psychological disorders such as anxiety and depression, or obvious brain lesions.	AUD severity (MAST score): 12.69 ± 4.12	Minimum (days): 14	IGT GDT	▪ IGT total net score: AUD < CG ▪ IGT net scores block 1–3: AUD = CG ▪ IGT net scores block 4–5: AUD < CG ▪ IGT learning rates: The performance of the CG improved over blocks, whilst the AUD group showed reduced net scores, particularly from block 3 onward. ▪ GDT net score: AUD < CG ▪ GDT single combination selection (risky option): AUD > CG ▪ GDT double combination selection (risky option): AUD = CG ▪ GDT triple and quadruple combination selection (safe options): AUD < CG ▪ GDT: the AUD group made more disadvantageous choices than the CG.	+
	**CG** (*n* = 50) Age (y): 47.70 ± 6 Sex (m/f): 43/7 Education (y): 10.72 ± 2.06						
Zorlu et al. (2013) [104]	**AUD^#^** (*n* = 17) Age (y): 47 ± 7 Sex: Male (all participants) Education (y): 7.6 ± 2.8	Exclusion criteria for the AUD group included other substance disorders, current or past history of Axis I psychiatric disorders (except for a past major depressive disorder), a current or past history of significant neurological disorders, or severe medical issues (e.g., HIV).	AUD duration (y): 12.2 ± 7.3 Duration of use (y): 29.7 ± 7.5 Age at first use (y): 17.3 ± 3.9 Quantity (drinks per day): 18.2 ± 3.7	Minimum (days): 14	IGT	▪ Total net score: AUD = CG ▪ Scores blocks 1–4: AUD = CG ▪ Scores block 5: AUD < CG ▪ Learning rates: the performance of the CG improved over blocks (particularly in the last two blocks), whilst the AUD group did not make the shift from disadvantageous to advantageous choices.	+/−
	**CG** (*n* = 16) Age (y): 46.7 ± 7.5 Sex: Male (all participants) Education (y): 8 ± 2.2						
Zorlu et al. (2014) [105]	**AUD^#^** (*n* = 12) Age (y): 53.9 ± 6.3 Sex: Male (all participants) Education (y): 7.6 ± 2.8	Past history of alcohol-induced mood disorder with depressive features: *n* = 7 Exclusion criteria for the AUD group included other substance disorders, current or past history of Axis I psychiatric disorders (except for a past alcohol-induced mood disorder with depressive features), a current or past history of significant neurological disorders, or severe medical issues (e.g., HIV).	AUD duration: 24.6 ± 9.8 Duration of use: 39.2 ± 6.9 Age at first use (y): 14.8 ± 3.7	Months: 27.8 ± 22.6 (range: 6–72)	IGT	▪ IGT total net score: AUD = CG ▪ Learning rates: AUD = CG ▪ Learning rates: the performance of the CG shifted from the advantageous to the disadvantageous decks over the last two blocks, whilst the AUD group did not make the shift from disadvantageous to advantageous choices.	−
	**CG** (*n* = 13) Age (y): 52.1 ± 4.4 Sex: Male (all participants) Education (y): 8 ± 2.2						
Zorick et al. (2022) [62]	**AUD^#^** (*n* = 12) Age (y): 43.2 ± 8.1 Sex (m/f): 8/4 Education (y): 12.9 ± 1.5	Depression (BDI score): AUD = CG Exclusion criteria for both groups included any recent (comorbid) mental illness, including depression.	Quantity (drinks per day): 6.0 ± 3.3	Study included active alcohol dependent individuals. Participants should maintain abstinence for a minimum of 16 h per testing day.	BART CFT/LA(G)T	▪ BART adjusted number of pumps: AUD = CG ▪ CFT/LA(G)T loss aversion: AUD > CG	−
	**CG** (*n* = 13) Age (y): 38.3 ± 8.9 Sex (m/f): 5/8 Education (y): 13.8 ± 1.5						

**Notes:** ^1^ Group characteristics are reported as mean ± standard deviation, unless otherwise indicated. AUD^#^ refers to a participant group meeting DSM-IV (TR)/ICD-10 criteria for alcohol abuse or alcohol dependence. AUD* refers to a participant group meeting DSM-V criteria for alcohol use disorder. ^2^ Refers to comorbidities/exclusion criteria in/for the AUD group(s) only, unless otherwise indicated. ^3^ Indicated for AUD group(s) only. ^4^ < and > refers to significant group differences at the alpha levels reported in the original studies. = refers to insignificant group differences at the alpha levels reported in the original studies. ^5^ This column includes an interpretation of the study outcomes regarding between-group comparisons of the task performance of the AUD groups and CGs. A‘+’ refers to a study predominantly having found significant between-group differences in the most important outcome measures related to risky decision-making of the included task(s). A ‘−’ refers to a study predominantly having found insignificant differences between the AUD group and the CG in the outcome measures related to risky decision-making. A ‘+/−’ refers to mixed results in that group differences were observed in some, but not all outcome measures related to risky decision-making, or on some, but not all included risky decision-making tasks. ^6^ At-risk drinkers were participants who answered yes to the question ‘in the past 12 months have you had three or more alcoholic drinks within a 3 h period on three or more occasions’, and reported on alcohol use at least once a week in the past year. ^7^ CD refers to a history of conduct disorder. ^8^ Different values were reported for the years of heavy drinking in the two studies by Noël et al. (2007; 2011) [54,55], as other demographic characteristics overlap completely; a typo was presumably made in one of the two studies. **^a^** Studies presumably made use of the same total AUD sample (*n* = 102). As the two studies did different sub-group analyses, they are reported separately in the table, but considered conjointly in the content and meta-analysis. **^b^** Studies presumably made use of the same total AUD sample (*n* = 48), and CG (*n* = 36). As the two studies did different sub-group analyses, they are reported separately in the table, but considered conjointly in the content and meta-analysis. **Abbreviations (in alphabetical order):** ADHD = Attention Deficit Hyperactivity Disorder. ADHD-RS = Attention Deficit Hyperactivity Disorder Rating Scale. ADS = Alcohol Dependence Scale. AIDS = acquired immunodeficiency syndrome. API = Addiction Profile Index. ASP(D) = antisocial personality (disorder). AUD = alcohol use disorder (group). AUD + A/C = alcohol use disorder group with ≥1 cluster A or C personality disorder. AUD + B = alcohol use disorder group with ≥1 cluster B personality disorder. AUD-HD = alcohol dependence high number (≥2) of detoxifications. AUD-LD = alcohol dependence low number (<2) of detoxifications. AUD-RA = alcohol dependence recently (≤16 days) abstinent. AUD-LA = alcohol dependence longer (>16 days) abstinent. AUDIT = Alcohol Use Disorders Identification Test. BART = Balloon Analogue Risk Task. BDHI = Buss-Durkee Hostility Inventory. BDI(-II) = Beck Depression Inventory(-II). BPRS = Brief Psychiatric Rating Scale. CD = conduct disorder. CFT/LA(G)T = Coin Flipping Task/Loss Aversion (Gambling) Task. CG = control group (without (severe) psychiatric and/or neurological disorders). CGT = Cambridge Gambling Task. CPT = Card Playing Task. CT = Cups Task. DSM = Diagnostic and Statistical Manual of Mental Disorders. EDMT = Ecological Decision-Making Task. EGT = Explicit Gambling Task. GDT = Game of Dice Task. h = (number of) hours. HADS: Hospital Anxiety and Depression Scale. HD = heroin dependence. HD + AUD = heroin dependent alcohol use disorder group. HIV = Human Immunodeficiency Virus. IGT = Iowa Gambling Task. IPV = interpersonal violence perpetrators. LRT = Lane Risk-taking Task. MADRS = Montgomery Asberg Depression Rating Scale. MAST = Michigan Alcohol Screening Test. MGT = Mixed Gambles Task. MWTB = Multiple Choice Vocabulary Test. m/f = male/female ratio. *n* = sample size. NEO = Neuroticism-Extraversion-Openness. n.r. = not reported. OCDS = Obsessive Compulsive Drinking Scale. PACS = Penn Alcohol Craving Scale. PD = Probability discounting. PDT = Probabilistic Discounting Task. TBI = Traumatic Brain Injury. T1 = time moment 1. T2 = time moment 2. T3 = time moment 3. RTT = Risk Taking Task. SADQ = Severity of Alcohol Dependence Questionnaire. SCID = Structured Clinical Interview for Mental Disorders. SD = standard deviation. SD-AUD = substance dependent AUD group. SOG = Single Outcome Gambling task. STAI = State-Trait Anxiety Inventory. STAXI = State-Trait Anger Expression Inventory. WoF = Wheel of Fortune task. y = years.

### 3.2. IGT

Thirty of the included studies made use of the IGT to assess risky decision-making in adults with AUD. In total, 25 studies indicated that the AUD group showed an aberrant performance (i.e., performed in a riskier fashion) as compared to the CG on one or more of the outcome measures used (25/30 = 83%). Specifically, 20 out of 28 studies (71%) reporting on the total net scores, total financial outcome, or total number of advantageous/disadvantageous choices indicated the overall task performance to be significantly worse (i.e., riskier) in the AUD group as compared to the CG [54,55,64,70,73,76,78,79,82,84,89,90,92,93,94,95,97,99,100,101,103]. This increased risk taking corresponds to the medium pooled effect size of 0.56 (95% CI = 0.45–0.68, *p* < 0.001) that was found in the subgroup meta-analysis for the IGT, which included 27 studies. A total of 18 studies compared the outcomes of the AUD groups and CGs on the individual task blocks of the IGT, 17 of which (94%) reported on significant group differences [54,55,61,64,70,78,79,81,86,89,90,93,94,95,97,101,103]. In these studies, adults with AUD were found to predominantly show an impaired performance as compared to the CGs in the last two to three blocks of the IGT. As the IGT is designed in such a way that the participant acquires knowledge of outcome probabilities as the task proceeds (i.e., across task blocks), performance in the first blocks is considered to reflect decision-making under ambiguity, whilst performance in the later blocks reflects decision-making under risk [106]. Thus, the impairments in the last IGT blocks are indicative of increased risk taking in the AUD groups. Looking at the learning rates across the five blocks specifically, 12 out of 17 studies (71%) found the learning rates to differ significantly between the AUD groups and the CGs [54,55,60,64,70,82,86,89,93,94,101,103,104]. Whilst the participants from the CGs generally learned to choose from the more advantageous/safe decks throughout the task, AUD participants, for example, showed no learning effect, showed a smaller improvement than participants from the CGs across blocks, or showed an initial improvement followed by a performance drop in the last block(s) of the task. In the remaining five studies that made use of the IGT (5/30 = 17%), no significant between-group differences were observed between the AUD groups and the CGs in overall task performance, performance on the individual blocks, or in the learning rates across the task [52,53,77,85,96,105].

Apart from making use of the original IGT, Kim et al. (2006) [64] also adopted a variant version of the IGT. The only difference between the two task versions is that in the variant version of the IGT, the order of punishment and reward of the decks is reversed. On the original IGT, Kim et al. (2006) [64] found significant group differences between the AUD groups and the CGs in overall performance, block performance, and learning rates. Contrastingly, no significant group differences were found regarding the overall performance (i.e., total net score) and the block scores on the variant version of the IGT. Kim et al. (2006) suggest that these mixed findings with regard to the IGT versions may be due to the AUD groups being less sensitive to punishment than the CGs, but equally sensitive to reward [64].

### 3.3. CGT

The nine studies that made use of the CGT provide inconsistent evidence of increased risk taking in adults with AUD as compared to the control participants. In total, four studies (4/9 = 44%) indicated that the AUD groups showed an aberrant performance as compared to the CGs on one or more of the different outcome measures related to risky decision-making [83,87,91,102]. The remaining five studies (5/9 = 56%) did not find significant between-group differences on the outcome measures related to risk taking of the CGT [48,49,50,51,77,80,88]. Including all nine studies that adopted this task, the subgroup meta-analysis for the CGT showed a small pooled effect size of 0.30 (95% CI = 0.13–0.47, *p* < 0.001).

### 3.4. BART

Four of the five studies (80%) that used the BART did not observe significant differences between the AUD groups and the CGs on the outcome measures related to risky decision-making [17,62,96,98]. In one of the five studies (20%), significant differences were observed between the AUD group with comorbid ADHD and the CG on the ‘total number of balloons that popped’, which was considered a measure of maladaptive risk taking. However, no significant differences were observed on this outcome measure between the AUD group without ADHD and the CG. Furthermore, no group differences were observed between the three study groups in the ‘adjusted number of pumps’, which was considered a measure of adaptive risk taking [81].

### 3.5. RTT

As described above, the studies by Bjork et al. (2008) [46] and Zhu et al. (2016) [47] were considered conjointly with regard to the overall conclusions because they made use of the same AUD sample and an overlapping control sample. This resulted in a total of two studies making use of the RTT to assess risky decision-making in adults with AUD [46,47,67]. Overall, the results of these studies provide inconsistent evidence of increased risk taking in adults with AUD as compared to the CGs. Whilst Bjork et al. (2004) [67] initially only found significant group differences in the trial maximum and total number of ‘busts’ on the RTT, the between-group differences became more pronounced when education level was entered as a covariate. When controlling for education level, the AUD group was shown to perform significantly worse (i.e., riskier) compared to the CG on all outcome measures of the RTT, including the trial average. Trial average was defined as the primary measure of risk taking [67]. In the study by Bjork et al. (2008) [46], significant group differences were only observed for the reward accrual time (risk-taking measure), and the number of ‘busts’ in the low-penalty trials. No significant group differences were observed in the high-penalty trials, or in the amount of money earned during the task, for example. In the study by Zhu et al. (2016) [47], who used the same AUD sample as Bjork et al. (2008) [46] but added 17 participants to the CG, no significant between-group differences were reported for any of the outcome measures of the RTT.

### 3.6. GDT

The two studies that made use of the GDT both found that the AUD participants selected the risky choice options significantly more frequently and the safe choice options significantly less frequently than the CGs. In both studies (100%), the AUD groups were found to show a significantly worse (i.e., riskier) overall task performance as compared to the CGs [89,103].

### 3.7. CFT/LA(G)T

Three studies adopted the CFT/LA(G)T as a measure of risky decision-making [34,62,70]. Two studies (2/3 = 67%) showed that the AUD group performed the task in a significantly riskier fashion than the CG. Brevers et al. (2014) [70] reported that, as compared to the CG, AUD participants displayed a significantly elevated acceptance to gamble in high-risk trials. In line with this finding, Genauck et al. (2017) [34] indicated that the AUD group showed a significantly reduced loss aversion and a significantly lower sensitivity to losses as compared to the CG. The reported sensitivity to gains did not differ significantly between groups [34]. In contrast to these two studies [34,70], Zorick et al. (2022) [62] (1/3 = 33%) found the loss aversion of the AUD group to be significantly greater as compared to the CG. According to Zorick et al. [62], one explanation for this opposite finding could be that loss aversion differs between active alcohol users as included in their study as compared to individuals with AUD who are currently abstinent (see [34,70]), suggesting a more direct effect of alcohol on task performance.

### 3.8. EDMT

Arcurio et al. (2015) and Folco et al. (2021) [20,45] both made use of the same AUD and control samples to describe different aspects/outcome measures of the EDMT. These studies were therefore considered conjointly with regard to the overall conclusions on differences between the AUD groups and CGs. Arcurio et al. (2015) [45] investigated risky decision-making in adults with AUD in the context of their decision to drink. The AUD group was found to endorse the high-risk alcohol stimuli significantly more frequently than the CG. No significant between-group differences were observed for the low-risk alcohol stimuli and for the high- or low-risk appetitive and neutral cues. In the context of risky sexual decision-making, and in line with the findings by Arcurio et al. (2015) [45], Folco et al. (2021) [20] found the AUD group to endorse high-risk sexual stimuli (faces) significantly more frequently than the CG. Again, no significant differences were observed between the AUD group and CG in the low-risk condition.

### 3.9. Outcomes on Other Risky Decision-Making Tasks

The EGT [58], CT [70], CPT [73], WoF [56], SOG [57], LRT [59], PDT [35], and MGT [35] were all used in one of the included studies only. Results of these studies are summarised in Table 2. Overall, significant differences in risky decision-making were observed between the AUD groups and CGs on five of these eight tasks (i.e., the EGT, CT, CPT, PDT, and MGT). Explanations of significant between-group differences, for example, included a lower motivation of the AUD group to perform the task than the CG [73], higher levels of impulsivity, and a reduced sensitivity to punishment in adults with AUD [35]. In contrast, Brevers et al. (2014) [70] suggested sensitivity to losses to be intact in adults with AUD, as they found the AUD group to show significantly increased risk taking in the gain domain but not in the loss domain of the CT. No significant between-group differences were found on risk-related outcome measures of the LRT, SOG, and WoF. On the WoF, Galandra et al. (2020) [56] did however find that adults with AUD displayed longer deliberation times and impaired adaptations of their decisions to previous feelings of regret and disappointment.

### 3.10. Associations between Risky Decision-Making and Demographic, Clinical, and Alcohol-Use-Related Variables

Several studies looked into associations between the outcome measures of the risky decision-making tasks and demographic, clinical, and/or alcohol-use-related variables in the AUD group(s) and/or CG(s) by means of relational analyses (e.g., correlational or regression analyses) or comparisons between different (AUD) groups. A summary of the findings regarding these associations is provided below.

#### 3.10.1. Demographic Characteristics

Of the eight studies that looked into the association between age and risky decision-making, three studies observed significant, although opposite, correlations between age and the outcome measures related to risk taking. In the AUD groups, a higher age was found to correlate significantly with a worse (i.e., riskier) performance on the CGT [83], and with a better (i.e., less risky) performance on the RTT [67] and the EGT [58]. Contrastingly, three studies found no significant correlation between age and IGT performance in the AUD group [76,94,99] or the CG [76]. In combined samples of AUD and control participants, two studies further found no significant correlations between age and IGT [70,82], CFT/LA(G)T [70], and CT performance [70].

Exploring the association between gender or sex (hereafter referred to as gender) on the risky decision-making tasks, five studies performed correlational analyses or an analysis of variance and did not find a significant association between gender and performance on the IGT [70,82,90,94], CGT [51], CFT/LA(G)T [70], and CT [70] in either a combined sample of AUD and control participants [51,70,82,90], or in the AUD group [94]. Similarly, four of the five studies that directly compared groups of male and female participants did not find significant gender differences in performance on the IGT [61,78,85], or in risk taking on the CGT [49]. Only Fein et al. (2004) [84] found that males showed significantly worse (i.e., riskier) IGT performances than females. 

Regarding the association between the performance on risky decision-making tasks and education level, three studies found that a higher level of education correlated significantly with a better performance on the IGT [82,89] or the RTT [67]. However, seven studies did not find significant correlations between performance on the IGT [70,76,86,94,99,101], CFT/LA(G)T [70], CT [70], or EGT [58] and the level of education of the AUD and/or control participants.

#### 3.10.2. Clinical Characteristics

None of the studies that looked into the association between symptoms of depression or anxiety and risky decision-making found significant correlations between the depression and/or anxiety ratings and performance on the IGT [52,70,73,94,99], CFT/LA(G)T [70], CT [70], or CPT [73]. Likewise, in the regression model introduced by Körner et al. (2015) [90], scores on the depression rating did not explain a significant proportion of the variance in IGT performance in the AUD group or CG. Furthermore, IGT and CPT performance did not correlate significantly with ADHD symptoms [73]. Finally, Brière et al. (2019) [78] found IGT performance not to be dependent on current psychotropic treatment, personal psychiatric history, or a family history of psychiatric problems.

Looking at the (additive) effects of psychiatric disorders comorbid to AUD, Dinesh et al. (2022) [81] found the AUD group with comorbid ADHD to perform significantly worse on the BART than the CG. As no significant difference was observed between the AUD group without ADHD and the CG on the BART, this finding suggests an (additive) effect of ADHD on maladaptive risk-taking behaviour. It should be noted, however, that this finding seems to be task specific. In the same study, no significant differences were observed between the AUD groups with and without ADHD and the CG on IGT performance [81]. Looking at the (additive) effects of personality disorders, Dom et al. (2006) [82] observed that adults with AUD and a comorbid cluster B personality disorder (i.e., borderline disorder, antisocial personality disorder, or both) performed worse (i.e., riskier) on the IGT than adults without a comorbid personality disorder or adults with AUD and a comorbid cluster A or C personality disorder. This suggests that the severity of impairments on the IGT relates to the presence of cluster B personality characteristics [82]. In line with this finding, Miranda et al. (2009) [95] showed that the AUD group with a comorbid antisocial personality disorder performed worse (i.e., riskier) in the last block of the IGT as compared to the AUD group without an antisocial personality disorder. Finally, in the study by Kim et al. (2006) [64], a history of conduct disorder comorbid to AUD was shown not to have an additive effect regarding the performance on the IGT or an IGT variant.

Regarding the relation between (co-)addictions and risky decision-making, Bernhardt et al. (2017) [35] observed a significant negative correlation between smoking status (i.e., smoking or non-smoking) and performance on the PDT. In contrast, Brevers et al. (2014) [70] found no significant association between IGT, CGT, or CT performance and the number of cigarettes consumed per day in a combined group of AUD and control participants. Similarly, two studies found no significant correlation between IGT performance and the smoking amount or history of the AUD group [76,99] or the CG [76]. In line with these findings, Loeber et al. (2009) [52] did not observe a significant correlation between the severity of nicotine dependence and IGT performance, and in the study by Brière et al. (2019) [78], no significant correlation was found between IGT performance and the presence of co-addictions. Finally, in the study by Murray et al. (2018) [96], no significant group differences were observed in IGT or BART performance between the smoking and non-smoking groups, and comorbid heroine dependence in the AUD group did not appear to have an additive effect on CGT performance in the study by Fischbein et al. (2007) [48].

#### 3.10.3. Alcohol-Use-Related Variables

Twenty studies investigated associations between alcohol-use-related variables and risky decision-making. Fifteen of these twenty studies did not find significant associations between the outcome measures related to risk taking and any of the alcohol-use-related variables that were studied. This included variables related to the frequency, quantity, or duration of alcohol consumption, ratings of AUD severity and/or alcohol cravings, and variables related to the duration of abstinence, or the number of prior detoxifications [34,35,53,58,64,67,78,82,85,89,90,93,94,97,101]. In total, only five of the twenty studies found alcohol-use-related variables to be significantly related with outcome measures of risky decision-making [52,54,76,84,99]. In two studies [76,99], a significant negative correlation was found between IGT performance and ratings of alcohol craving. Contrastingly, three of the included studies did not find significant correlations between ratings of alcohol craving and risky decision-making task performance [34,35,64]. In the study by Fein et al. (2004) [84] a significant negative correlation was found between IGT performance and the duration of peak alcohol use. However, in their study from 2006, Fein et al. [85] did not find IGT performance and the duration of peak drinking to correlate significantly. Noël et al. (2007) [54] further found that the group of AUD participants who showed an impaired performance on the IGT had a higher daily dose of alcohol use, a higher number of prior detoxifications, and a longer duration of AUD as compared to the group of AUD participants who showed an unimpaired performance. In contrast, various included studies did not find significant associations between risky decision-making task performance and quantity-related measures [35,85,93,94], the prior number of detoxifications [53,64,78,93,94], and/or the duration of AUD [78,82,93,94]. Finally, Loeber et al. (2009) [52] found the IGT performance of AUD participants who were recently abstinent (i.e., ≤16 days) to be worse (i.e., riskier) as compared to the performance of AUD participants with a longer abstinence period (i.e., >16 days). In the same study, participants with a higher number of detoxifications (i.e., ≥2) furthermore showed significantly lower learning rates across the IGT blocks than the group with a lower number of prior detoxifications (i.e., <2). However, in nine other studies, no significant associations were found between task performance and the duration of abstinence [60,78,84,97,101], or the prior number of detoxifications [53,64,78,93,94].

##### Relapse to Drinking

Four of the included studies specifically looked at the relation between risky decision-making and relapse to heavy drinking [35,53,77,94]. Three of these four studies (75%) found task performance and relapse to be significantly related. In the context of a longitudinal study, Loeber et al. (2010) [53], for example, found the IGT net score at baseline to be a significant predictor for relapse in the AUD groups. Notably, participants with a higher net score (i.e., less risky performance) at the first assessment during inpatient treatment were more likely to relapse within the first six months after discharge. In both regression models introduced by Bernhardt et al. (2017) [35] (see Table 2 for an overview of included variables), on the other hand, a lower probability discounting for losses on the PDT (i.e., riskier performance) was a significant predictor for a greater hazard of relapse during the 48-week follow-up interval. Similarly, two of the three studies that compared abstaining and relapsing AUD groups found the relapsing AUD groups to perform significantly worse (i.e., riskier) than the abstaining groups on the PDT [35], IGT [77], and CGT [77]. Contrastingly, Maurage et al. (2018) [94] (1/4 = 25%) observed no significant group differences between the abstaining and relapsing AUD groups in IGT performance.

## 4. Discussion

Referring to a maladaptive pattern of alcohol intake characterized by the inability to stop or control alcohol consumption despite its detrimental consequences [8], AUD forms a major health concern and is the most common substance use disorder worldwide [10]. In the literature, the behavioural and cognitive deficits associated with AUD have often been related to deficits in risky decision-making. Importantly, these decision-making deficits have been suggested to be both a consequence of and a risk factor for the maintenance of alcohol use [22]. Whereas both theoretical and behavioural research seem to have established a link between increased risk taking and AUD, no extensive literature review and meta-analysis has been performed to date that focuses on risky decision-making in adults with AUD, specifically. The objective of the present study was therefore to provide a comprehensive overview and meta-analysis of the existing studies that compare the performance on risky decision-making tasks of an AUD group with the performance of a CG. This study therewith aimed to examine the magnitude and type of deficits in risky decision-making of adults with AUD and to explore the potential mechanisms behind these deficits.

In total, 50 studies were included in the present review that presumably made use of distinct (i.e., non-overlapping) participant samples. Overall, 68% of the included studies reported that the AUD group(s) showed an aberrant performance as compared to the CG(s) on one or more of the adopted risky decision-making task(s), as was indicated by significant between-group differences on one or more of the outcome measures used. In line with this finding, a small to medium pooled effect size of 0.45 was found in a global meta-analysis comparing the level of risk taking of the AUD groups and the CGs. Consistent with previous theoretical and behavioural research [7,23,24,25], the present review provides evidence of increased risk taking in adults with AUD as compared to control participants. In regard of this overall conclusion, it should be noted, however, that for 30% of the included studies, no significant between-group differences in risky decision-making were found, and that in one study (2%), outcomes on one of the two adopted tasks were indicative of reduced risk taking in adults with AUD as compared to the CG [62]. Furthermore, the pooled effect size in the global meta-analysis could be interpreted as small to medium only. This latter finding may be explained by a smaller number of studies being included in the meta-analysis than in the qualitative synthesis, and by differences in the tasks and outcome measures for risky decision-making considered for the analyses. Apart from outcomes on the adopted risky decision-making tasks, potential mechanisms behind the identified deficits in risky decision-making will be explored in light of the SMT of addiction. Further points for discussion include the associations between risky decision-making and demographic, clinical, and alcohol-use-related variables, the limitations of the present review, and, finally, recommendations for future directions of research.

### 4.1. Outcomes on Risky Decision-Making Tasks

Despite a lack of clarity about the construct validity and reliability of the instrument [107], a prominent finding of the present review is that the IGT appears to be largely overrepresented in the current evidence base as compared to other risky decision-making tasks. Indeed, 60% of the included studies made use of the IGT to assess risky decision-making in adults with AUD. In line with a previous study by Kovács et al. (2017) [108], reviewing the performance of adults with AUD or gambling disorder on the IGT, the vast majority of these studies indicated that adults with AUD performed in a riskier fashion as compared to a CG on one or more of the outcome measures of the IGT. The significance of these differences between the AUD groups and CGs was confirmed by a medium pooled effect size found in the subgroup meta-analysis for the IGT. Contrastingly, the subgroup meta-analysis for the CGT, which was used in 18% of the included studies, revealed a small pooled effect size only. This corresponds to the finding that, of the studies that made use of the CGT, only a minority (i.e., 44%) found significant group differences between the AUD groups and CGs in risky decision-making. The results of the present review therefore seem to indicate that the IGT appears more sensitive to the deficits in risky decision-making of adults with AUD than the CGT. In this context, it should be noted, however, that the confidence intervals of the pooled effect sizes for the IGT and CGT overlap slightly, and that the difference in effect size between the two tasks is statistically insignificant. As three of the studies that made use of the CGT were among the five largest studies included in this review (i.e., studies with the largest sample sizes), the use of this task will, nevertheless, have made a relatively large contribution to the small to medium effect size found in the global meta-analysis.

In spite of its relatively high sensitivity to detect decision-making deficits, the specificity of the IGT remains too low to identify the mechanisms or impairments underlying (deficits in) risky decision-making [89]. Due to the complexity of the task design of the IGT, multiple affective and cognitive processes seem to be involved when performing the task [70]. These processes are difficult to disentangle [89]. Whereas the IGT was originally designed as a task that mainly taps into affective decision-making (see [63,109]), several studies have shown that more cognitively demanding or deliberative processes may also be involved in IGT performance [70,106]. Specifically, the first blocks of the IGT have been suggested to reflect decision-making under ambiguity [106], relying more heavily on affective processes, as the probabilities of reward and loss of each deck are unknown [70]. As the task proceeds, however, participants can acquire knowledge of the outcome probabilities, and executive functions such as working memory, response inhibition, and cognitive flexibility may become more involved [54,70,106]. The later blocks of the IGT are therefore considered to reflect decision-making under risk rather than ambiguity. As such, an aberrant IGT performance in adults with AUD as compared to control participants may reflect impairments in both affective and deliberative decision-making processes (see [106,110]).

In contrast to the IGT, the CGT more clearly addresses decision-making under risk as the outcome probabilities of reward and loss are made explicit to the participants at the start of the task. The CGT is therefore typically described as a cognitive or deliberative decision-making task. This is not to say, however, that affective processes are not engaged when performing the task [111,112]. Indeed, results of imaging studies that have looked at the brain regions involved in CGT and IGT performance seem to suggest that the neural components associated with affective decision-making overlap with those associated with the deliberative decision-making required for the CGT [112].

Similar to the IGT and CGT, most of the other included risky decision-making tasks to some extent rely on both affective and deliberative processes. Overall, the outcomes of these decision-making tasks thence seem to be in support of the presence of deficits in both types of decision-making in adults with AUD. Accounting for deficits in affective and deliberative decision-making processes [3,4], the SMT of addiction may provide an explanatory framework for the association between deficits in risky decision-making and AUD as substantiated by the present review.

### 4.2. The SMT of Addiction

The SMT proposes that the decision-making process is guided by emotional or ‘somatic’ signals brought about by two interacting neural systems. The impulsive system, typically associated with affective decision-making, responds to environmental stimuli indicative of immediate rewards or pleasure, and activates feelings related to the immediate prospect of a decision. The reflective system, which is associated with deliberative decision-making, exerts a certain level of control over the impulsive system and can activate feelings related to the future prospects of a choice. The somatic signals triggered by both systems compete until one signal prevails. This signal consequently guides the decision to be made. In the context of this theory, AUD can be understood as an imbalance between the two neural systems, emerging either from a hyperactive impulsive system or from a dysfunctional reflective system [23,24,25].

In general, it can be concluded that the deficits in risky decision-making observed in the majority of included studies (i.e., 68%) are consistent with the SMT of addiction. As significant between-group differences were found on predominantly affective as well as deliberative decision-making tasks or outcome measures, the findings of the present review furthermore support the idea of both a hyperactive impulsive and a dysfunctional reflective system in adults with AUD. Indeed, in line with the idea of increased activation of the impulsive system, AUD participants displayed more risky choice behaviour than the CGs in the majority of included studies. Being associated with affective decision-making in particular, hyperactivity of the impulsive system mainly becomes apparent during task aspects that reflect decision-making under ambiguity, where task rules are less explicit. Due to altered functions such as a reduced sensitivity to losses or punishment (i.e., reduced learning from previous decisions and feedback) [34,35,64], for example, the impulsivity to make riskier choices may be less easily repressed, which in turn aligns with the idea of a defective reflective system. In regulating the impulsive system, the reflective system is particularly thought to be dependent on brain regions associated with executive functioning (e.g., regions of the prefrontal and cingulate cortex) [24]. Impairments of adults with AUD in working memory, response inhibition, or cognitive flexibility, for example, could form an explanation for the observed deficits on the deliberative aspects of the included decision-making tasks (e.g., later trials of the IGT and CGT performance).

When exploring the potential mechanisms behind deficits in risky decision-making, it should also be noted, however, that it remains unclear whether the imbalance between the impulsive and reflective system predates the addiction of the AUD participants, or whether it is consequential to their alcohol use (see [24,25]). Moreover, the finding of the present review that 30% of the included studies reported no significant differences between the CGs and adults with AUD questions the applicability of the SMT as a comprehensive explanatory framework. As described above, the absence of significant results in these studies possibly relates to the sensitivity of the chosen task to detect risky decision-making deficits, or could perhaps be explained by differences between the studies in demographic, clinical, or alcohol-use-related variables.

### 4.3. Associations between Risky Decision-Making and Demographic, Clinical, and Alcohol-Use-Related Variables

Apart from looking at differences between the AUD groups and CGs, several of the included studies explored potential associations between risky decision-making and demographic, clinical, and/or alcohol-use-related variables within groups. With regard to the demographic characteristics age and educational level, results of the included studies were mixed in that some but not all studies observed significant associations. The direction of the association between risky decision-making and age furthermore remains unclear, as a higher age was found to correlate both with a worse/riskier [83] and with a better/less risky [58,67] performance on the tasks. Results were more consistent regarding the association between risk taking and gender, since only one of the ten studies that looked at this relation found a significant effect of gender on decision-making task performance [84]. Despite the mixed and contrasting results regarding demographic characteristics, the potential influence on task performance of age and educational level in particular cannot be ruled out. Future studies that assess risk-taking behaviour in adults with AUD should therefore aim to control for between-group differences in demographic variables.

When looking at the relations between clinical characteristics and risky decision-making, only a few significant associations have been found in the included studies. Apart from a potential additive effect of ADHD [81], or a cluster B personality disorder [82,95], the results of the present review seem to suggest that risk taking in adults with AUD is not affected by clinical characteristics such as symptoms of depression and anxiety or co-addictions to nicotine or heroine. However, in this context, it is of note that the included studies differed with regard to the inclusion and exclusion of adults with disorders comorbid to AUD. For many of the included studies, adults with (severe) psychiatric comorbidities and/or polysubstance dependency were excluded from the AUD group in order to address the ‘pure’ effect of AUD on task performance. As AUD is associated with various psychiatric comorbidities including mood, anxiety, and personality disorders [11], excluding participants with comorbid disorders may limit the generalizability and ecological validity of the study results. Based on the included studies, it therefore remains unclear whether associations with clinical characteristics are also absent in the population of adults with AUD.

Similar to the absence of associations with clinical characteristics, the results of the present review indicate risky decision-making in adults with AUD not to be associated with a variety of alcohol-use-related measures. Indeed, the majority of included studies that looked at these associations found that outcome measures of risky decision-making were not related to the frequency, quantity, or duration of alcohol consumption, to ratings of AUD severity and/or alcohol cravings, or to the duration of abstinence, or the number of prior detoxifications in the AUD groups. As both co-addictions and alcohol-use-related variables seem to be of little influence on the risk taking of the AUD groups, it could be speculated that the deficits in risky decision-making substantiated by this review are not a direct consequence of the alcohol or substance use itself, but rather of the brain deficiencies related to the addiction. Importantly, such brain deficiencies could both predate the addiction and be involved in the maintenance of alcohol use [22]. In support of the idea that risky decision-making relates to the maintenance of alcohol use, three of the included studies found risky decision-making task performance to be related to a relapse to heavy drinking in abstinent adults with AUD [35,53,77]. In this context, it should be noted, however, that the direction of the association between risk taking and relapse remains unclear, based on this limited number of studies. Whilst two of the included studies found a riskier task performance to be related to relapse [35,77], the study by Loeber et al. (2010) found a less risky task performance to relate to a relapse to heavy drinking. According to the authors, this latter finding could be due to the impact of social desirability on task performance (i.e., a safety strategy is adopted during the task), however, and might not generalise from the laboratory setting to real-world risk situations [53]. Future research will therefore have to show whether the finding by Loeber et al. (2010) was incidental, and will need to clarify the relation between risk taking and relapse. Establishing a relation between the deficits in risky decision-making of adults with AUD and a relapse to heavy drinking may in turn prove helpful for the development of new treatment approaches in the light of relapse prevention.

### 4.4. Limitations and Future Research Directions

The results of this systematic and meta-analytic review must be interpreted with caution and several limitations need to be acknowledged. A first limitation concerns the ecological validity of the present findings. Whereas the present review indicates that adults with AUD show increased risk taking as compared to control participants on performance-based decision-making tasks, relatively little can be said, based on these tasks, about their level of risk taking in daily life, and about their decisions in relation to alcohol consumption in particular. Whereas the IGT, for example, was designed to assess real-world decision-making [63], and has been linked to clinically relevant risk-taking behaviour such as pathological gambling, psychopathic behaviour, and substance use disorders, further evidence on the ecological validity of this task is still needed [107]. Factors relevant to real-world risky decision-making including social context, emotional arousal, and the extent of punishment and reward do not translate easily to a controlled and structured laboratory setting [113]. Furthermore, whilst tasks that include decisions about points or money such as the IGT and CGT may shed light on risky decision-making processes in a broader sense, their relevance to alcohol-related decision-making remains to be explored [114]. In order to more firmly establish the link between task performance and real-world risk taking in adults with AUD, future research should aim to develop and adopt more ecologically valid risky decision-making tasks that specifically relate to the decision to drink alcohol. One example of such a task included in the present review may be the EDMT as introduced by Arcurio et al. (2015) [45].

As described above, a second limitation includes that, based on the present findings, it remains unclear whether the deficits in risky decision-making substantiated by this review predate the addiction of adults with AUD, and/or whether the deficits are consequential to their alcohol use. This question of causality can only be answered by means of a longitudinal approach, and is of clear relevance for the early identification of those at risk for the development of addiction and for enhancing treatment options for adults with AUD. To gain a further understanding of the (bidirectional) nature of the link between risky decision-making and AUD, it is thus highly recommended that more longitudinal research is conducted on this topic, particularly in the clinical field [22].

A third limitation of this review is the high level of variability of the included studies regarding the level of comorbidities, the alcohol-use-related variables (e.g., duration and severity of AUD), and the abstinence periods of the AUD groups. This heterogeneity between studies complicates their comparibility for the present review. Moreover, differences in the sample characteristics and the characteristics related to the alcohol comsumption of the AUD groups could differentially impact risky decision-making behaviour, which may consequently have influenced the results.

A fourth limitation that manifests itself on the review level concerns that the present study was not preregistered. For reasons of transparency and reproducibility, registration of the study protocol should have been completed.

A final limitation on the review level is that no risk of bias analysis has been carried out for the present study. After careful evaluation of the assessment tools and checklists at hand (see [115]), we concluded that, to the best of our knowledge, there is no tool available to date that is fully applicable and can be used to reliably address the risk of bias of the type of cross-sectional studies included in the present review (i.e., studies comparing behavioural task performance between a clinical group and a CG). Therefore, we were unable to examine the likelihood that features of the study design or conduct of the included studies have led to bias, which should be taken into account when interpreting the results of this review.

## 5. Conclusions

This systematic and meta-analytic review provides evidence of increased risk taking in adults with AUD as compared to control participants. Specifically, the majority of studies (i.e., 68%) reported that the AUD group(s) showed an aberrant performance as compared to the CG(s) on one or more of the adopted risky decision-making task(s), which was confirmed by a small to medium pooled effect size found in the meta-analysis. In the present review, significant between-group differences were found on predominantly affective as well as deliberative decision-making tasks or outcome measures. Accounting for deficits in affective and deliberative processes, the overall pattern of findings of this review thence seems to be consistent with the SMT of addiction. In the context of this theory, AUD can be understood as an imbalance between two neural systems, emerging from a hyperactive impulsive system or a dysfunctional reflective system [23,24,25]. The applicability of the SMT as a comprehensive explanatory framework is questioned, however, by the finding that 30% of the included studies reported no significant group differences in risky decision-making between the CGs and adults with AUD.

Apart from looking at differences between the AUD groups and CGs, potential associations between risky decision-making tasks and demographic, clinical, and alcohol-use-related variables were explored. Whereas results regarding associations with demographic characteristics were mixed, the majority of clinical and alcohol-use-related variables under review were found not to be significantly related with risk taking in adults with AUD. As (the absence of) such associations may hold relevant implications for both theory and practice (e.g., relapse prevention), future research should aim to systematically evaluate the relation between these variables and risky decision-making in adults with AUD. It is further recommended that future studies aim to adopt a longitudinal approach, and make use of more ecologically valid tasks that specifically relate to the decision to drink alcohol. Information yielded by this type of research can teach us more about the role of real-world risky decision-making in the development and maintenance of AUD.

The present systematic and meta-analytic review is an important step in adding to our understanding of the link between risky decision-making and alcohol addiction. The outcomes of this review may prove useful for relapse prevention and the development of new treatment approaches for adults with AUD.

## Figures and Tables

**Figure 2 jcm-12-02943-f002:**
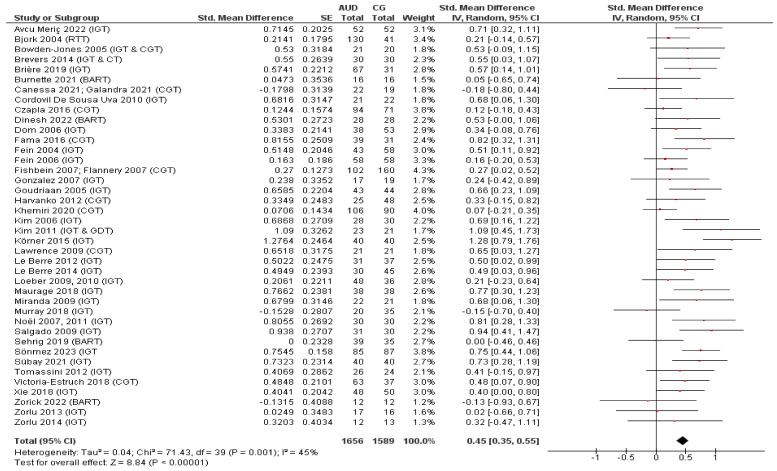
Forest plot of studies assessing the level of risk taking of adults with AUD compared to control participants included in the global meta-analysis. **Note:** The black diamond represents the pooled result of the studies included in the global meta-analysis (pooled Hedges’ *g* = 0.45, 95% CI = 0.35–0.55). **Abbreviations (in alphabetical order):** AUD = alcohol use disorder (group). BART = Balloon Analogue Risk Task. CG = control group (without (severe) psychiatric and/or neurological disorders). CGT = Cambridge Gambling Task. CI = confidence interval. CT = Cups Task. df = degrees of freedom. GDT = Game of Dice Task. IGT = Iowa Gambling Task. IV = weighted mean difference. P = probability value. RTT = Risk Taking Task. SE = Standard Error. Std. Mean Difference = Standardized Mean Difference.

**Figure 3 jcm-12-02943-f003:**
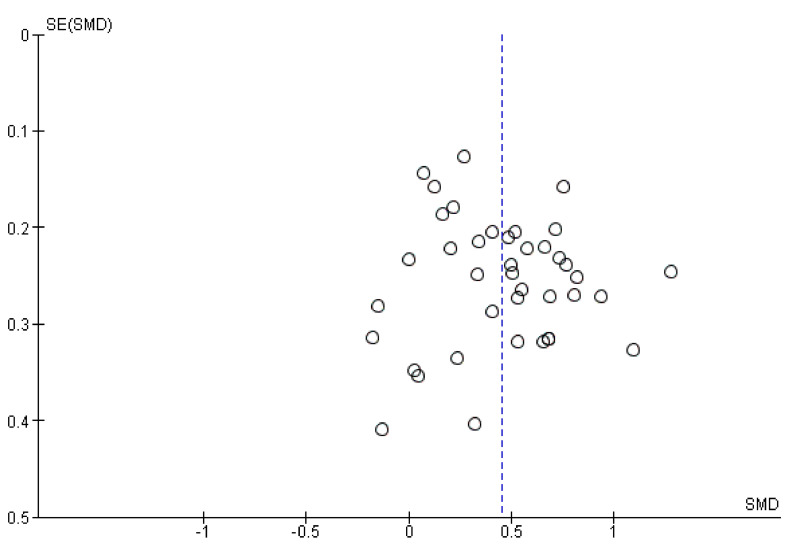
Funnel plot of studies assessing the level of risk taking of adults with AUD compared to control participants included in the global meta-analysis. **Note:** The circles represent the individual studies included in the global meta-analysis, the blue line represents the overall effect (pooled Hedges’ *g* = 0.45). An asymmetric funnel plot indicates publication bias. **Abbreviations (in alphabetical order)**: SE = Standard Error. SMD = Standardized Mean Difference.

## Data Availability

The data that support the findings of this study are available on request from the corresponding author.

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
