# Peer review of "Risky Decision-Making in Adults with Alcohol Use Disorder—A Systematic and Meta-Analytic Review"

_jcm, 2023, doi:10.3390/jcm12082943_

Round 1

Reviewer 1 Report

The characteristics of this work result in extensive work that is hard to follow. The document becomes more readable if the authors could be more succinct and clearer.

I also consider fundamental an update of search.

Abstract

The Goal and conclusions of the abstract should be more clear. It’s seem confusing.

 Introduction

 Indeed, adults with AUD appear more likely to engage in (health-related) risk behaviour than healthy individuals (e.g. [19–21]) – Why e.g.?

What is the biggest difference between your work and work from Chen et al. (2020)?

The authors could increase the pertinence of this study, whether they indicate what the possible results from this study may contribute to society.

In general, the introduction section should be more clear.

 Methods

The authors used adequated methodology to this type of work. However, I suggest na updated to Prisma to 2020 PRISMA (prisma-statement.org).

Journal articles were searched using the databases PsycINFO, MEDLINE, PubMed, and Web of Science. Why these databases?

The search was carried out on April 2021, almost 2 years gone. An update of this search is needed.

Study analysis – How authors solved when there was disagreement between researchers in the exclusion or inclusion of studies?

Meta analysis – the authors used mean difference or standard mean difference?

Risk Bias was assessed?

Result/Discussion

This section is hard to follow. Please reconsider rewriting some paragraphers.

Conclusion

It is important to reinforce the impact of the results in practice and the novelty of this work.

Reviewer 2 Report

Dear Authors,

It is a well conducted Systematic and Meta-Analytic Review following standardized methodology according to PRISMA guidelines, with clear and specific study question, specific eligibility/exclusion criteria for included studies, appropriate/comprehensive search strategy with the aim of including all relevant studies, data abstraction, using appropriate statistical techniques, evaluating the results, assessment of heterogeneity and publication bias and appropriate conclusion.

Minor recommendations:

1. You did not perform quality assessment of individual studies with standardized tools, although you explain this in limitation section. Did you reject the Newcastle Ottawa Scale (NOS) (Wells et al., 2018) for cross-sectional studies? Also, you did not register your review’s protocol (in PROSPERO).

2. On page 41 under the heading ‘Relapse To Drinking’, you mention: Participants with a higher net score (i.e., less risky performance) at the first assessment during inpatient treatment were more likely to relapse within the first six months after discharge, Loeber et al. (2010). This article on discussion section states: ‘However, in the present study, our results indicated that a lower net outcome was associated with a lower frequency of relapse. Thus, patients following a safety strategy by preferring cards with lower immediate gains but fewer losses were more likely to relapse’. Maybe you could elaborate more on this result.

3.  On page 3 under the heading ‘Study Selection Procedure’ you mention that studies were only included when they (a) compared a group of adults with AUD as their main clinical diagnosis to a CG without (current) psychiatric or neurological disorders…But for example you included Gonzalez et al. (2007) study with AUD participants with current major depressive disorder: 71%. It is not the only included with psychiatric commorbidities (depression and anxiety), for ecological validity and generalizability purposes. On page 45 you state: ‘In this context it is of note, however, that the included studies differed with regard to the inclusion and exclusion of adults with disorders comorbid to AUD. For many of the included studies, adults with (severe) psychiatric comorbidities and/or polysubstance dependency were excluded from the AUD group in order to address the ‘pure’ effect of AUD on task performance’. So, in your eligibility/exclusion criteria instead of referring to the absence of (current) psychiatric or neurological disorders it is better to refer to the absence of (severe) psychiatric or neurological disorders.

4. Since you mention that AUD participants with comorbid cluster B personality disorder, performed worse (i.e., riskier) on the IGT compared to the AUD group without such a personality disorder, why did you include studies from Dom et al. 2006 and Miranda et al. 2009, in your meta- analysis?

5. Abstinence Period in many studies is not reported and in some others is very short (24 hours etc). Doesn’t this affect results? Should this be included in limitation section?

6. Page 5 on results section: ‘For the global meta-analysis, five of the fifty included studies were excluded from the because they did not report on a risk-related outcome measure’. Please revise.

7. In several spots throughout the text, like on the first page: ‘This so-called ‘risky decision making’ involves intuitive as well as deliberative thought processes (e.g. [3,4])’, you give examples stating the references, without concluding the sentence or explaining, making it difficult to follow. Please revise.    

Round 2

Reviewer 1 Report

Thank you by addressing my comments. 

It is important to indicate a methodology to studies inclusion criteria, like PICOS per example.

Why protocol wasn't registered? It is author's option. This needs a justification.
